

# Introducing the Probabilistic Earth-System Model: Examining The Impact of Stochasticity in EC-Earth v3.2

Kristian Strommen[1], Hannah M. Christensen[1], Dave MacLeod[1], Stephan Juricke[2,3], and Tim N. Palmer[1]

[1]Department of Atmospheric, Oceanic and Planetary Physics, Oxford University, Oxford, UK.
[2]Alfred Wegener Institute, Helmholtz Centre for Polar and Marine Research, Bremerhaven, Germany.
[3]Jacobs University Bremen, Bremen, Germany.

**Correspondence:** Kristian Strommen (kristian.strommen@physics.ox.ac.uk)

**Abstract.** We introduce and study the impact of three stochastic schemes in the EC-Earth climate model, two atmospheric schemes and one stochastic land scheme. These form the basis for a probabilistic earth-system model in atmosphere-only mode. Stochastic parametrisations have become standard in several operational weather-forecasting models, in particular due to their beneficial impact on model spread. In recent years, stochastic schemes in the atmospheric component of a model have been shown to improve aspects important for the models long-term climate, such as ENSO, North Atlantic weather regimes and the Indian monsoon. Stochasticity in the land-component has been shown to improve variability of soil processes and improve the representation of heatwaves over Europe. However, the raw impact of such schemes on the model mean is less well studied, It is shown that the inclusion of all three schemes notably change the model mean state. While many of the impacts are beneficial, some are too large in amplitude, leading to large changes in the model's energy budget. This implies that in order to keep the benefits of stochastic physics without shifting the mean state too far from observations, a full re-tuning of the model will typically be required.

## 1 Introduction

One of the key guiding principles of the scientific method is the need to assess and quantify uncertainty. The truncation of the true climate system to a finite grid necessarily introduces a large source of uncertainty from unresolved sub-grid scale processes. While parametrisations are usually developed to account for these unresolved processes (Bauer et al., 2015), the parametrisation process relies on introducing a number of simplifications and assumptions that are not always valid, effectively introducing an additional layer of uncertainty in any model prediction (Palmer et al., 2005). Some of these assumptions are resolution-dependent: for example, convection parametrisations typically assume that the size of a grid-box is large enough for the grid box to contain a large sample of clouds, such that the average influence of the clouds is well constrained by the resolved flow (Arakawa and Schubert, 1974; Lord et al., 1982). As model resolution continues to increase, such assumptions can become increasingly tentative, even while the resolution is still far from being explicitly cloud resolving. The need to represent the uncertainty of the sub-grid contribution to the flow therefore becomes increasingly important.

In medium-range and seasonal forecasts using numerical weather prediction models, the use of stochastic schemes has become widespread as a means to sample this uncertainty. Studies have shown that when properly calibrated, such schemes

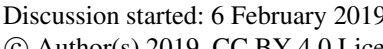

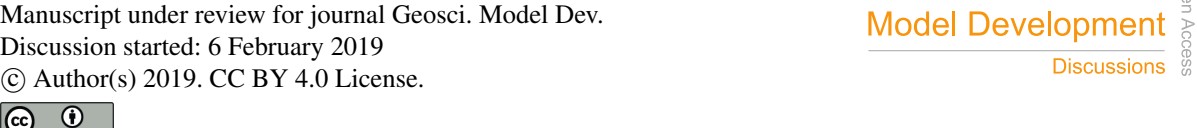

have a beneficial impact on both the spread and mean state of these forecasts (Weisheimer et al., 2011; Berner et al., 2017; Leutbecher et al., 2017). In recent years, there has been increasing interest in understanding the impact of these schemes also on the long-term climate of a model. In (Palmer, 2012), it was argued that introducing stochasticity into climate models may be a key step towards eliminating persistent model biases and reducing uncertainty in climate projections. Since then, the insertion

of a stochastic component into a climate model has been demonstrated to improve the several key processes, including El Niño-Southern Oscillation (Christensen et al., 2017a; Yang et al., 2018; Berner et al., 2018), the Madden Julian Oscillation (Wang and Zhang, 2016) and the representation of the Indian monsoon (Strømmen et al., 2018). Improvements were also found on regime behaviour, northern hemispheric blocking patterns and tropical precipitation (Dawson and Palmer, 2015; Davini et al., 2017; Watson et al., 2017). Most of these studies focused on a particular, multiplicative noise scheme called the 'stochastically

perturbed parametrisation tendencies' scheme (SPPT). A more flexible variant of this scheme (dubbed 'ISPPT') was developed and found to substantially improve the skill of weather forecasts in areas with significant convective activity, though only a limited evaluation of its impact on longer timescales was reported (Christensen et al., 2017b). However, modern climate models incorporate also a full land-system and are coupled to an ocean model, both of which carry their own sources of uncertainties. In (Macleod et al., 2016), stochasticity was added to the land-scheme of the IFS, and was found to have a positive impact

on seasonal predictability, as well as the 2003 European heatwave. In the ocean, a number of schemes have been considered, including perturbations of ocean mixing processes and sea ice (Juricke et al., 2013; Juricke and Jung, 2014; Juricke et al., 2017, 2018). Both variability and mean-states of key quantities were found to improve. There have more recently also been some studies focusing on the impact of stochastic schemes on the ocean-atmosphere coupling (Williams, 2012; Rackow and Juricke, 2019).

The idea behind the 'Probabilistic Earth-System', as put forward in (Palmer, 2012), is to incorporate stochastic representations of model uncertainty not only into the atmosphere, but also the land, sea ice, and ocean components of the EC-Earth model, thus obtaining a state-of-the-art earth-system model with stochasticity in each major component. Such a fully probabilistic coupled climate model will be tested in the PRIMAVERA project (Haarsma et al., 2016). In this paper, we will present the schemes used in the land and atmosphere components, and perform initial tests in an atmosphere-only configuration. This

allows us to test the raw impact of the atmosphere and land schemes on the mean climate with no ocean coupling. In this sense, the study conducted here is a first test of the configurations to be used in the PRIMAVERA simulations. A key motivating question is whether the benefits of such schemes can be achieved without any additional tuning.

As will be seen, the two atmosphere schemes and one land scheme have a notable impact on the energy budget of the model, implying that in coupled mode, the inclusion of different combinations of these schemes can be expected to shift the

model climate to a notably different stable state. In fact, even in these atmosphere-only simulations, we demonstrate significant changes in the mean state of several variables on the large scale, such as atmospheric water vapour content, cloud liquid water, cloud cover, soil temperature and soil moisture. This highlights that the non-linear impacts of random model error, as represented through a zero-mean stochastic perturbation, cannot be neglected in climate models. While some of the changes are positive compared to reanalysis data, not all are. In particular, key global quantities such as net surface energy that models are





frequently tuned for, can be strongly altered. This implies that while adding stochasticity can be beneficial, additional model tuning may be required to keep the mean state close to observations.

In section 2, we describe the model used, and the stochastic schemes under consideration. In section 3, we describe the experiments carried out, reference data and statistical methods. Section 4-7 contains the actual analysis. We choose to focus our evaluation on the impact of these stochastic schemes on the mean climate. In sections 4-6, we examine, for each scheme in turn, changes in the global mean of key variables relative to a set of deterministic simulations. These changes provide insight into the impact of the schemes on the energy budget and the hydrological budget. In section 7 we compare changes in the modeled circulation as represented by the Hadley cell and quasi-biennial oscillation (QBO) across all three schemes (SPPT, ISPPT and Stochastic Land). Finally, section 8 contains a discussion on the cause of the observed changes, as well as our conclusions.

We note that while we do not test the stochastic ocean schemes here, basic tests of the impact of these on the ocean component NEMO of EC-Earth can be found in (Juricke et al., 2017) and (Juricke et al., 2018).

## 2 Model Description

### 2.1 About EC-Earth

EC-Earth is an Earth system model developed by the international EC-Earth consortium (Hazeleger et al., 2012). The atmospheric component uses a modified version of the Integrated Forecast System (IFS) used by the European Centre for Medium-Range Weather Forecasts (ECMWF). Land-surface processes are simulated using the Hydrology Tiled ECMWF Scheme of Surface Exchanges over Land (H-TESSEL) (Balsamo et al., 2009). More details are given in section 2.3. Dynamic ocean coupling is available using the Nucleus for European Modelling of the Ocean (NEMO) model version 3.6. The coupling in this case is handled with OASIS3 (Valcke, 2013).

The tests in this paper were performed using EC-Earth version 3.2.1, a stable version for atmosphere-only simulations, based on cycle 36r4 of the IFS (released in 2010). The IFS solves the resolved processes in spectral space, with advection and physics parametrisations computed on a reduced Gaussian grid. The parametrizations used are described extensively in (Beljaars, 2004). Modifications to the IFS are carried out to improve the model performance on climate time-scales. In particular, various parameters controlling the physics parametrizations were tuned by the consortium to obtain a realistic energy budget in the period 1990-2010, as compared to observational estimates (Trenberth et al., 2009).

In all experiments carried out, we used the default resolution of the current EC-Earth version, which uses 91 vertical levels and a spectral resolution of T255, corresponding to a horizontal resolution of around 80 kilometres.

### 2.2 Description of the SPPT and ISPPT schemes

The SPPT scheme has been included in the operational model of the IFS since 1998. The scheme is designed to represent forecast uncertainty that arises from the simplifications and approximations involved in the parametrisation of unresolved




atmospheric processes. It does this by perturbing the total net tendency from the physics parametrisations using a multiplicative noise term $r$:

$$\hat{\mathbf{P}_{\mathbf{k}}} = (1 + \mu_k r) \sum_{i=1}^{6} \mathbf{P}_{i,k},$$  (1)

where $\mathbf{P}_{i,k}$ is the tendency vector for the prognostic model variables (winds, temperature and humidity) from the $i$-th physics

parametrisation scheme, where $k$ indicates the vertical model level. The perturbation, $r$, is constant in the vertical, but is tapered through $\mu_k \in [0, 1]$ to smoothly reduce the perturbation to zero in the boundary layer and stratosphere. The perturbation $r$ follows a Gaussian distribution with mean zero, and is smoothly correlated in space and time. The implementation in EC-Earth follows that in the Integrated Forecasting System as described in (Palmer et al., 2009). The perturbation $r$ is generated by summing over three independent spectral patterns with standard deviations (0.52, 0.18, 0.06), spatial correlation lengths (500

km, 1000 km, 2000 km) and temporal decorrelation scales (6 hours, 3 days, 30 days) respectively. The perturbation $r$ is limited between zero and two to ensure $\hat{\mathbf{P}}$ has the same sign as $\mathbf{P} = \sum \mathbf{P}_i$.

The SPPT scheme described above assumes that the different physics parametrisations have the same model error characteristics. If the net tendency is small, SPPT represents the associated model uncertainty as small, even if the associated individual tendencies were large. (Christensen et al., 2017b) proposed a generalisation to the SPPT approach in which each physics

process is perturbed independently using multiplicative noise:

$$\hat{\mathbf{P}} = \sum_{i=1}^{6} (1 + \mu_k r_i) \mathbf{P}_{i,k},$$  (2)

The random patterns, $r_i$, are independently generated and evolved. The statistical properties of the $r_i$ (standard deviation, spatial and temporal correlations) can be individually specified to account for differences in the uncertainty characteristics arising from each physics scheme. This generalisation, 'Independent SPPT' (ISPPT), was shown to significantly improve the

reliability of ensemble forecasts in the tropics, and in areas with significant convective activity (Christensen et al., 2017b). This indicates ISPPT is likely a better representation of the uncertainty associated with convection than SPPT. Further justification for the use of ISPPT over SPPT has been provided by considering coarse-graining experiments (Christensen, 2018).

### 2.3 Description of the Stochastic Land scheme

The land surface model used here is H-TESSEL, the Tiled ECMWF Scheme for Surface Exchanges over Land (TESSEL)

with revised land surface hydrology, comprising a surface tiling scheme and vertically discretized soil. The surface tiling scheme allows each gridbox a time-varying fractional cover of up to six tiles over land (bare ground, low and high vegetation, intercepted water, and shaded and exposed snow) and two over water (open and frozen water). Each tile has a separate energy and water balance, which is solved and then combined to give a total tendency for the gridbox, weighted by the fractional cover. Full details of the model may be found in (Balsamo et al., 2009) for details.

Stochasticity is induced in the land-scheme via the hydraulic conductivity, which is calculated in H-TESSEL using the van Genuchten formulation (van Genuchten, 1980). This formulation is favoured by soil scientists as it has shown good agreement



with observations in intercomparison studies (Shao and Irannejad, 1999). The hydraulic conductivity is in this formulation given by

$$\gamma = \gamma_{sat} \frac{[(1+\alpha h^n)^{1-1/n} - \alpha h^{n-1})]^2}{(1+\alpha h^n)^{(1-1/n)(l+2)}}, \tag{3}$$

where $\alpha$, $l$ and $n$ are soil texture parameters, dependent on the soil type, $h$ is soil water potential (the potential energy of
soil water due to hydrostatic pressure), and $\gamma_{sat}$ the hydraulic conductivity at saturation. The two key soil parameters $\alpha$ and
$\gamma_{sat}$ of Eq. 3 are stochastically perturbed. These particular parameters have been chosen as previous studies found them to be
particularly sensitive (Cloke et al., 2008), (MacLeod et al., 2016). Each parameter is perturbed with its own integration of the
spectral pattern generator. As with SPPT and ISPPT, the spectral pattern is obtained as a sum of three patterns, each with the
same spatial and temporal decorrelation scales as for those schemes. The weightings for each scale is in this case set to 0.33.
The two parameters are observed to have some correlation (Cosby et al., 1984) and so a third pattern is used as a base for each
pattern, in order to introduce correlation between them. The correlation coefficient between the two parameters is prescribed
to be 0.6 on average. The perturbation is multiplicative, as with SPPT. That is, each parameter $p$ is multiplied by $(1+r)$ where
$r$ is the randomly generated spectral pattern described above.

Previous work has investigated the impact of representating uncertainty in these particular hydrology parameters. Perturbing
coupled atmosphere-ocean seasonal hindcast experiments, (Macleod et al., 2016) resulted in an improved representation of soil-
moisture driven processes active during the 2003 heatwave, leading to an improved signal in the seasonal forecast of surface
air temperature. In a set of atmosphere-only experiments, (MacLeod et al., 2016) showed a strong sensitivity of soil moisture
memory to uncertainty in these parameters. Using the same atmosphere-only model, (Orth et al., 2016) also demonstrate an
improvement in subseasonal forecast skill through incorporation of land surface parameter uncertainty.

## 3   Data and Methods

### 3.1   Experimental setup and reference data

We performed four ensemble experiments, which we will for brevity refer to as CTRL, SPPT, ISPPT and LAND, referring
to the scheme being used in each (in the CTRL experiments no stochastic schemes were used). The ensemble members were
run for 20 years each, starting from February 1st of the following years: 1960, 1965, 1970, 1975, 1980. Thus the total period
covered by all the experiments is 1960-2000. By spacing out the members in this way we account for the possibility that the
impact of the schemes may be sensitive to the ocean state. Atmospheric initial conditions for these dates were provided by the
Climate Prediction group at the Barcelona Supercomputing Center. These initial condition files are obtained by interpolation
of reanalysis data using the FULLPOS post-processing software for the IFS: this carries out the interpolation using the model
executable to ensure minimal model drift (Bellprat et al., 2016). The sea surface temperatures and sea-ice are then specified
using the CMIP6 forcing datasets to match observations. Radiative forcings such as those from anthropogenic or volcanic
sources were provided by the same source.



For the SPPT scheme, we used the default settings for the magnitudes and time/length-scales of the perturbations as in (Leutbecher et al., 2017), described in section 2.2.

For ISPPT, we used the same settings for the 3 fields as for SPPT. We set the convection and large-scale water-process tendencies to share the same perturbation, and let the remaining four tendencies (radiation, turbulence and gravity wave-drag and non-orographic gravity wave drag) have separate, independent perturbations. Following the naming convention introduced in (Christensen et al., 2017a), this corresponds to ISPPT {1,2,3,3,4,5}, where the number indicates which random seed was used to generate a pattern and the ordering of the numbers indicates which of the 6 physics parametrisation schemes are perturbed with that pattern: 1st=radiation (RDTN), 2nd=turbulence, vertical mixing and orographic drag (TGWD), 3rd=convection (CONV), 4th=large scale water (cloud) processes (LSWP), 5th=non-orographic drag (NOGW), and 6th=methane oxidation (MOXI).

For the LAND scheme, the parameter perturbations $r$ were restricted to be strictly between 1 and -1, as numbers greater than or equal to 1 were found to produce unphysical runoff behaviour. For this reason, the standard deviation of the noise perturbations were set to 0.33, allowing three standard deviations to explore the full available range.

To allow for a single reanalysis dataset to be used across all the experiments, ERA40 (Uppala et al., 2005) was chosen as our reference point. When evaluating the QBO, we use the more recent reanalysis dataset ERA-Interim (Dee et al., 2011).

## 3.2 Methodology

### 3.3 Statistical Techniques

All fields considered were filtered by taking monthly means. To compute the mean impact of the scheme, each simulation is paired with the corresponding CTRL simulation covering the same time period. The difference of each individual pair is computed, and the mean across the five ensemble pairs gives the mean impact of the scheme. When comparing the CTRL simulations with reanalysis, each CTRL simulation is paired with the reanalysis data for the corresponding time-period, and again the mean across all five pairs defines the CTRL bias. The standard deviation across the five ensemble members is used to estimate the statistical significance of the mean differences between model simulations. Twice the standard deviations is used as an error bound (displayed either with shading in time-series plots or explicitly written as a number in percentage change plots). Because there are only five samples, it is hard to claim that differences are definitively significant even if the zero-line is further than two standard deviations away from the mean. However, in the cases where this does happen, we have observed that all five individual differences have the same sign, suggesting that in these cases the difference is likely significant.

To assess differences in spatial patterns, the five ensemble members are concatenated to produce a timeseries spanning 1200 months at each grid point for each simulation, with the ensemble members ordered according to their start date. A two-tailed T-test is applied to assess the significance of the difference in the mean.Spatial plots then indiate the mean difference across all the five simulations. Gridpoints where the difference lies outside the 95% confidence interval are marked with dots. A cruder test was also used which simply tested if four of the five differences had the same sign. The results were almost identical.

Finally, to asses whether a reduction in the mean-square error (MSE), relative to re-analysis, was significant, we computed the MSE for all five simulations of the scheme in question, as well as the five MSE's of the CTRL simulations. If the difference





between the mean CTRL MSE and the mean MSE of the scheme was greater than two standard deviations of the spread in MSE's across the five scheme simulations, the difference was deemed significant.

## 3.4 Energy Budget

As the simulations were performed with historical forcings, the energy budgets of the five CTRL simulations are not compara-

ble with each other. However, since the EC-Earth Consortium tuned the deterministic model to have a realistic energy budget over the period 1991-2010, the surface fluxes of the CTRL simulation covering the period 1991-2000 are shown in Table 1. The table has been split into two parts, as the Pinatubo eruption in 1991 has a strong influence on the first half of the period. For reference, the estimates of these quantities from (Trenberth et al., 2009) (covering 2000-2004) are also shown.

It can be seen that the CTRL simulation achieves a reasonable energy balance, with the net surface energy close to the

observational estimate. As will be seen, the adjustment to this budget by the introduction of the different stochastic schemes is very fast and approximately constant in time. Therefore, it is possible to assess whether the stochastic schemes lead to an improved or degraded energy budget overall, though it should be kept in mind that variations in the budget between years is often large.

## 4 The impact of SPPT

### 4.1 Impact on mean state

Figure 1(a) shows the percentage changes in global mean quantities due to SPPT, relative to the CTRL simulation. Evaporation has notably increased by about 1%. Figure 3(b) shows the spatial distribution of these changes[1] averaged across all seasons, to be compared against the bias of the CTRL simulations to reanalysis, Fig. 3(a). The increase can be seen to be concentrated in the western Pacific oceans in particular, with a notable decrease in the eastern Pacific. We note that the spatial changes due

to SPPT are not well correlated with the CTRL bias, reducing the bias in places and exacerbating it elsewhere. As the overall mean CTRL bias is close to 0, the net increase in evaporation due to SPPT represents a small degradation. This is also reflected in the mean-square error (MSE) between the spatial means of SPPT and reanalysis, as found in Table 2, where it can be seen that the MSE has increased with SPPT.

By contrast, precipitation, as seen in Fig. 4(b), has generally improved. This can be seen both in terms of the mean bias,

which has approximately halved, and the MSE (Table 2). While the precipitation biases of the CTRL model (plot (a) of the same figure) are targeted well in key areas, such as the Pacific and Indian oceans, the scheme exacerbates the biases elsewhere, such as over the maritime continent and in sub-Saharan Africa.

The scheme has also notably affected clouds, with cloud cover decreasing at all levels and cloud liquid water robustly increasing. Figure 7(b) shows the spatial distribution of total cloud cover changes. The spatial pattern of cloud liquid water

changes (not shown) are well correlated with these cloud cover changes, suggesting that while cloud cover in total has gone

---

[1]Note that we have not adhered to the IFS convention here that downward fluxes are positive. Thus red values indicate an increase in evaporation *upwards*.





down, the remaining clouds are more optically thick. Comparing Fig. 7(a) and (b) also shows that the net impact of SPPT on total cloud cover is to significantly reduce the CTRL bias. Indeed, SPPT has almost uniformly reduced the bias everywhere, bringing the mean bias down to nearly 0. The MSE has also gone down by about 15%.

The impact on the spatial biases of two meter temperature across all seasons are also shown in Fig. 6(b) The cold bias in the CTRL simulations (plot (a) of the same figure) across the equator can be seen to be robustly decreased by SPPT. The statistically significant increase in temperatures over sub-Saharan Africa are on the other hand introducing a bias that was not present in the CTRL model. No notable impact is made on the warm bias in the Arctic and Antarctic regions, and there is no notable change in the MSE overall, nor the total mean bias.

### 4.2 Impact on energy budget

Figure 8(a) shows the mean difference between each SPPT-simulation with the corresponding CTRL-simulation for the globally averaged surface energy fluxes shown in 1. Note that the standard EC-Earth convention has been used whereby downward fluxes are positive. Hence, a positive difference for a given flux indicates that the scheme increased the flux in the downwards direction on average, while a negative difference indicates an increase in the upwards direction on average. The benefit of adhering to this convention is that the net surface energy, always in black, can be obtained by summing up all the other flux-quantities in the plots. In this way, it is easy to assess which quantity is having the strongest influence on altering the net budget. The timeseries have also been smoothed by a 12-month running mean both to remove both the seasonal cycle and to highlight the overall trends.

It can be seen that the dominant impact of SPPT is a large increase in latent heat flux consistent with the strongly increased evaporation identified in the previous section. This in turn is the main contribution to the reduced net surface energy, with the total mean difference being -0.8 $W/m^2$. Compared to the baseline budget from Table 1, this clearly represents an unrealistic reduction in surface energy.

There is also a notable increase in surface solar radiation (SSR). The previous section indicated that, while cloud liquid water increased by around 1.2%, the total cloud cover decreased at all levels through the atmosphere. Thus it appears that while the clouds are more optically thick, which would tend to decrease SSR, the reduction in cloud cover leads to an overall increase in shortwave radiation reaching the surface.

### 4.3 Impact on the hydrological budget

Figure 2(a) shows the percentage changes in the total column soil water content (SWVLTOT) along with the three quantities responsible for modulating this quantity: precipitation, evaporation and total runoff. Note that precipitation and evaporation here have been restricted to land-points only, explaining why the numbers do not exactly match those of Fig. 1(a). There is a small but significant decrease in the total soil water of around 1%. As the SPPT scheme only directly interacts with atmospheric processes, this change will be driven by the observed changes in precipitation and evaporation. Over land, precipitation decreases by around 1.7% and evaporation increases by about 1%, both of which contribute to the decrease in soil water. The





notable decrease in runoff is likely a consequence of the reduction in water in the soil column. By reference to Table 2, the runoff changes, seen in Fig. 5(b) are beneficial overall, reducing the MSE.

## 5   The impact of ISPPT

### 5.1   Impact on mean state

Figure 1(b) shows the percentage changes in global means induced by the ISPPT scheme. The signal of increased evaporation and cloud liquid water seen with SPPT is greatly amplified. Figure 3(c) shows the spatial distribution of evaporation changes, which as with SPPT are strongest over the tropical oceans. The total increase in evaporation is even greater than that from SPPT, and again represents an overall increase in the global mean bias. However, there is an enchanced spatial coherence between the local changes from ISPPT and the CTRL bias (plot (a) of the same figure). It can be seen that the sign of the

ISPPT changes are generally opposite to the sign of the CTRL bias, indicating that the scheme is effectively targeting the local biases, with a notable exception being the ocean region south of Australia. This is reflected in the fact that while the global mean bias has increased, the MSE (Table 2) has decreased.

Figure 4(c) shows the changes in precipitation. Again, local biases are well targetted, with the MSE decreasing even more than with SPPT. The mean bias relative to reanalysis has been reduced to near 0.

Unlike SPPT, total cloud cover has only decreased by a very small amount, due entirely to a decrease in the high level clouds which dominates a small increase in low and mid-level clouds. Figure 7(c) shows the change in total cloud cover, where areas of overall decrease indicate decreases in high level cloud cover. As with SPPT, the changes serve to decrease the CTRL bias, both in total and locally, as can be seen by a notable decrease in the MSE (Table 2). Interestingly, there is great spatial coherence between these changes and those seen for SPPT in Fig. 7(b). Indeed, an examination of these and other variables

generally suggests that a first-order approximation of the *mean* impact of ISPPT is a stronger amplitude version of the impact of SPPT, but with the changes being even more closely correlated with the local biases, as seen with evaporation and precipitation above. The added freedom of non-correlated perturbations with ISPPT allows for a stronger influence on non-linear climate processes, most notably those associated with convective processes (Christensen et al., 2017b).

Figure 6(c) shows the local changes in two meter temperature. As with SPPT, the cold bias along the equator is reduced, but

unlike SPPT, the warm biases in the Arctic and Antarctic regions are also reduced. The globally averaged mean bias has been reduced by nearly 70%, and the MSE has decreased by nearly 25%.

### 5.2   Impact on energy budget

Figure 8(b) shows the flux impact of ISPPT. As with SPPT, the dominant effect is from the substantial increase in evaporation, as seen in Fig. 3(c). In contrast to SPPT, ISPPT also results in a substantial reduction in SSR of almost $1 \ W/m^2$. Figure 1(b)

shows that while there was a small decrease in total cloud cover, low level cloud cover (the layer reflecting the most solar radiation), has increased by around 0.5%. Even more notably, the cloud liquid water content has increased by nearly 5%,





increasing the albedo of these clouds significantly. These factors combine to explain the decreased SSR. A mild increase in sensible heat flux and thermal radiation cannot compensate this, such that the net result is a large decrease in surface energy of around 1.7 $W/m^2$ relative to CTRL. By reference to Table 1, this represents a large divergence from observed values.

### 5.3 Impact on hydrological budget

Figure 2(b) shows percentage changes for the hydrological budget for the ISPPT scheme. There is no notable change in precipitation over land, but evaporation has increased by around 3%. This will have a drying effect on the top soil layer. Despite this, no meaningful change is seen in the total column soil water. One possible explanation for this is to first note that in general, if the top soil layer holds more water, heavy rainfall events. will tend to saturate the surface, triggering the land-scheme to expel a lot of this water as runoff already at the top soil layer. This will tend to inhibit moisture from sinking

down to the lower soil layers. Conversely, if the top layer is drier, as seen with ISPPT (due to e.g. the increased evaporation), runoff at the top soil layer will not be triggered as easily during rainfall, allowing more water to reach the lower layers. In this way, increased evaporation can be balanced by reduced runoff to produce no overall change in total soil water. Noting the reduced runoff in Fig. 2(b), seen visually in Fig. 5(c), we speculate that this is the reason there is no change in soil water due to ISPPT. Of relevance to this is the fact that more frequent heavy rainfall events may be expected with ISPPT, as this is the case

with SPPT ((Watson et al., 2017)).

We finally note that by Table 2, the runoff changes are broadly beneficial, reducing the MSE even more than SPPT does.

## 6 The impact of Stochastic Land on the mean state

### 6.1 Impact on basic mean state

Figure 1(c) shows the percentage changes in global means induced by LAND. As with both SPPT and ISPPT, evaporation

increases. Spatial changes here are shown in Fig. 3(d), where visual inspection shows that the changes mostly serve to reduce the CTRL bias (plot (a) of the same figure). This is confirmed by a reduction in MSE, as seen in Table 2. As we will see in the section on hydrology, the changes in evaporation over land are strongly correlated with changes in runoff. Since evaporation changes over land can have knock-on effects on the global circulation more broadly, this already suggests a mechanism whereby the LAND scheme, which only directly interacts with soil processes, can still lead to global mean state changes.

For precipitation, seen in Fig. 4(d) and Table 2, while the MSE has gone down compared to the CTRL runs, the mean bias remains unchanged. While the precipitation bias over the Indian ocean and maritime continent are decreased, biases over India are increased and the changes over the Pacific ocean are fairly ambiguous in their merit.

Unlike SPPT and ISPPT, the LAND scheme has a more notable impact on total column water, increasing it by nearly 1%. The cloud cover changes are similar in characteristic to those of ISPPT, with low and mid-level cloud cover increasing, and

high level cloud cover decreasing: Fig. 7(d) shows the spatial distribution of these changes. It can be seen that while the LAND scheme has reduced biases over all the major continents (with the exception of Australia), as well as over the Indian Ocean, it





has had only a limited impact over the Pacific and Atlantic ocean. Consequently, the MSE, recorded in Table 2 has not changed relative to CTRL.

Figure 6(d) shows the spatial changes in two meter temperature, where the major, statistically robust change is a big reduction in the warm bias in the Arctic and Antarctic regions. This has reduced the overall mean bias relative to reanalysis by around
90%, as well as notably reducing the MSE.

## 6.2  Impact on the energy budget

Turning on the LAND scheme has only a small impact on the energy budget (Fig. 8(c)), increasing net surface energy by about $0.1 W/m^2$ relative to CTRL. The primary culprit appears to be a decrease in outgoing longwave radiation. Figure 1(c) shows that atmospheric water vapour has increased by about 1%, which would, by strengthening the greenhouse effect, serve to trap
more thermal radiation in the atmosphere. While the same figure shows high level cloud cover has decreased, which would tend to negate that effect, the large increase in mid level cloud cover may counteract the change in high level cloud to a large degree, such that the water vapour increase ends up being the dominant driver of energy budget changes.

## 6.3  Impact on the hydrological budget

Figure 2(c) shows the percentage changes observed on including the LAND scheme, restricted to land-points only. The scheme
has a big impact on both runoff and soil moisture over Antarctica. However, the run-off scheme is typically not behaving realistically here, where the soil is trapped under thick layers of ice. To avoid the quantified impacts being dramatically skewed from this contribution, we excluded Antarctica from the data prior to computing percentage changes[2].

Firstly, it can be seen that there is no statistically robust change to the overall soil water content. This is puzzling at first sight, since it can also be seen that there is an increase of precipitation over land which is not cancelled out by an equal amount
of evaporation; runoff has also decreased robustly by around 1%. These changes would be expected to lead to an increase in soil water. Indeed, examining the spatial distribution of soil water changes (not shown), one finds that areas of increased precipitation (Fig. 4(d)) do mostly correspond to areas of increased soil water content, and vice versa. The two main exceptions are over Greenland and Siberia, where a behaviour similar to that seen in Antarctica is observed (strongly decreased soil water and runoff) even with no meaningful change in precipitation. Figure 5(d) shows the spatial distribution of runoff changes. The
areas of increased runoff correlate well with areas of increased precipitation/soil water content, with runoff triggered more frequently the wetter the soil. The exceptions of Greenland and Siberia are clearly visible.

This suggests that in regions not dominated by ice and snow, long-term changes in runoff and soil water are driven by the long-term circulation changes induced when stochastically perturbing hydraulic conductivity. However, in regions such as Siberia and Greenland, there is a sharp decrease in soil water content within the first month of each LAND simulation, with
no associated change in surface evaporation. This extra decrease in soil water, independent of precipitation and evaporation changes, explains why the total soil water mean has not changed overall.

---

[2]Note that this was not done for the other schemes, where quantities computed globally also included Antarctica.



Comparing the runoff impacts with the CTRL bias (plot (a) and (d) of Fig. 5), we see that the local biases have mostly been improved, with the MSE (Table 2) decreasing by about 10%, suggesting the increased variability in the LAND scheme is serving to reduce the model bias.

## 7  Impact on atmospheric circulation

We now assess the impact of all three schemes on two key components of the atmospheric circulation: the Hadley Cell and the quasi-biennial oscillation.

### 7.1  Impact on the Hadley Cell

The impact of the stochastic perturbations on the atmospheric circulation is considered through analysis of the Hadley circulation. The Hadley cell varies with the seasonal cycle, and is stronger and wider in the winter hemisphere: we consider the
characteristics of the dominant Hadley cell separately for the key seasons December/ January/ February (DJF) and June/ July/ August (JJA). We characterise the zonally-averaged overturning circulation using a streamfunction, $\psi$, defined as a function of latitude and height, following (Waliser et al., 1999). The procedure is performed separately for each simulation.

To consider the effect of stochastic physics, we calculate three summary diagnostics for the cell. Firstly, the strength of the overturning is characterised using the maximum of the streamfunction. The width of the overturning cell is indicated by
estimating the latitude of the upwelling and downwelling branches of the cell respectively. This is defined as the latitude at which the streamfunction changes sign at the 700 hPa level. The 700 hPa level was chosen as this corresponds to the approximate level at which the streamfunction maximum is found. These results are shown for each ensemble member in Fig. 9 by the five scattered points. The diagnostics are also shown for ERA40 for the five time periods corresponding to the five ensemble members.

Panels (a) and (b) show the strength of the overturning circulation for each simulation and both seasons. The SPPT scheme shows no significant impact on the strength of the overturning in DJF, and a slight though not significant weakening in JJA. In contrast, both the ISPPT and LAND approaches show a significant strengthening of the overturning cell in both DJF and JJA. For all simulations, the JJA cell is stronger than the DJF cell. While the CTRL and SPPT simulations have a somewhat too weak Hadley cell, the Hadley cell in the ISPPT and LAND experiments is too strong.

Figure 9 (c) and (d) show the latitude of the downwelling branch of the dominant cell for each season. As for the strength of the circulation, SPPT has no significant impact on this diagnostic. Both the ISPPT and LAND schemes lead to a significant equator-ward shift of the downwelling latitude in DJF, in contrast to what is seen in reanalysis. In JJA all simulations are in agreement with ERA40, and only the LAND scheme leads to a slight equatorward shift.

Panels (e) and (f) show the latitude of the upwelling branch of the circulation. SPPT leads to no significant change in
this diagnostic, while the ISPPT and LAND schemes lead to a large shift poleward. This introduces a substantial bias when compared to ERA40, which needs to be understood. The net effect of this is to increase the width of the Hadley circulation in the ISPPT and LAND simulations.



The Hadley circulation is changing in response to climate change. There is evidence that it has widened over recent decades (Hu and Fu, 2007), (Seidel and Randel, 2007), and some evidence that it has also strengthened (Seager et al., 2007), though GCMs struggle to reproduce the observational signal (Mitas and Clement, 2005), (Johanson and Fu, 2009). To compare the climate change impact on the Hadley circulation for each stochastic model, we indicate the years covered by each simulation in Fig. 9.

The deterministic simulations show a general strengthening of the DJF Hadley circulation, though a weakening of the JJA Haldey circulation. This trend is also observed in DJF for the three stochastic models, though the signal in JJA is more mixed. While there is no climate change signal observed in the latitude of the downwelling branch for any simulation, the simulations generally agree that the DJF upwelling branch has shifted poleward. The exception is the ISPPT simulation, which does not show a strong sensitivity in this diagnostic.

## 7.2 Impact on the quasi-biennial oscillation

The QBO is a periodic downward propagation of easterly and westerly wind regimes which accounts for the majority of variability in the equatorial stratosphere (Baldwin et al., 2001), exerting a notable influence also on the extratropical atmosphere through modulating extratropical waves. Its period is typically estimated to be around 28 months.

Figure 10 shows the average QBO period across each simulation for each of the four experimental set-ups, as well as that for ERA-Interim. The period was here diagnosed using zonal equatorial (10S-10N) winds at 50hPa, zonally averaged. This produces a periodic timeseries from which we can readily estimate the average spacing between zero-crossings to determine the average period. When applied to the reanalysis dataset ERA-Interim, this produces a period of almost exactly 28 months, suggesting that this method captures the expected period well. It can be seen that the deterministic model itself cannot achieve nearly as long a period, falling just below 21 months. The stochastic schemes tend to make this slightly smaller, but compared to the initial bias of the deterministic model itself these changes are small.

Figure 11 shows a time-pressure level section of monthly averaged zonal wind, restricted to the equatorial region 10S-10N with the seasonal mean removed. This gives the usual visual representation of the QBO for the ensemble members covering the 1980-2000 period, with ERA-Interim over the same period also shown. It can be seen that indeed EC-Earth struggles to attain both a strength, period, and extent of downwelling comparable to ERA-Interim. None of the stochastic schemes notably change this.

## 8 Discussion and Conclusions

### 8.1 Discussion

A detailed investigation of the underlying causes of the changes documented above would require higher frequency temporal output than is available from the experiments considered here, as the changes observed appear to be firmly in place within





the first month of each experiment. Such an investigation will therefore be left to future work. However, for completeness, we include some discussion here on what the key processes at play may be.

With both SPPT and ISPPT, the dominant impact on the energy budget is increased evaporation. Evaporation in the IFS is controlled primarily by the wind-speed and humidity gradient near the surface (see (ECMWF, 2017)). While wind-speeds do increase by about 1.4% on average with ISPPT, the mean wind-speeds are unchanged with SPPT, with a tiny increase of only 0.06%. Given that the increase in evaporation of both SPPT and ISPPT are of the same order of magnitude, this suggests changes in the humidity gradient are a key factor. One possibility is that the increase in cloud liquid water is steepening the near-surface humidity gradient, causing more favourable conditions for evaporation. The fact that both cloud liquid water and surface wind-speeds increase more with ISPPT would then explain why this impact is amplified in those experiments. Furthermore, the increase in cloud liquid water with both SPPT and ISPPT could be due to an asymmetric response to stochastic perturbations in the convection schemes. Given a parcel of air close to saturation, whether the model actually triggers condensation depends sensitively on the humidity and temperature tendencies, both of which are perturbed by the stochastic schemes. A perturbation in one direction may result in condensation, and thereby an increase in the cloud liquid water, while a perturbation in the other direction leaves the parcel stable and the total cloud liquid water unchanged. Since the stochastic perturbations are zero mean, it is expected that these two scenarios will occur at the same rate, such that the net impact of the perturbations is to increase the total amount of cloud liquid water, as observed.

We suggest that the changes in evaporation and cloud liquid water are the main sources of large-scale changes to the mean climate caused by SPPT and ISPPT. These variables strongly control cloud formation, cloud albedo and latent heat release, which are the dominant sources of changes to the energy budget. In addition, changes to evaporation (and thereby precipitation) dominate the hydrological budget. Since the atmospheric circulation is also coupled to thermodynamic processes, especially in the tropics, it is likely that the observed impact on the Hadley cell can also be traced to these changes in the hydrological cycle. This is supported by (Numaguti, 1993), which showed that the strength and meridional structure of the Hadley cell is closely linked to the distribution of evaporation. With both ISPPT and LAND the change in the Hadley cell is largely deterimental compared to reanalysis, implying that some of the positive impacts seen on the mean state may be due to a compensation of errors. This would need to be studied more carefully in future work.

For the LAND scheme, the first-order impact appears to be regional changes in average runoff. That such changes should be expected to occur can be understood by reference to Eq. (3) defining hydraulic conductivity. Because runoff is triggered when the soil becomes saturated, its triggering is intimately linked to the ratio $\gamma/\gamma_{sat}$, which reaches its maximum of 1 precisely at saturation. This ratio, as seen in Eq. (3), is highly non-linear in the perturbed parameter $\alpha$, implying that even mean-zero perturbations can be expected to alter the mean state. This will lead to regional changes in the moisture content of the soil layer, which in turn influences evaporation over land. The net impact is a decrease in runoff and therefore an increase in evaporation. This change permeates through to influence the rest of the climate system, including an increase in total column water and large changes in the vertical distribution of cloud coverage.




## 8.2 Conclusions

Three stochastic schemes are introduced into the atmosphere and land components of the EC-Earth climate model. The stochastic schemes incorporate zero-mean perturbations into the model physics to represent uncertainty associated with unresolved, sub-grid scale variability. The interaction of these perturbations with the non-linear Earth-system results in systematic changes

to the mean state of the model in a way that is not a priori obvious. Schemes that are fairly similar (SPPT and ISPPT) may have very different impacts, and schemes that are only directly affecting a relatively small component of the model (LAND) may still notably change the global circulation. This highlights the importance of representing random model error in climate models, as well as in initialised simulations, where stochastic schemes have long been used to improve the reliability of forecasts (Palmer et al., 2009),(Berner et al., 2017).

Our experiments showed that the inclusion of all three stochastic schemes, particularly ISPPT and LAND, led to notable reductions in model biases compared to the deterministic model. This is seen perhaps most strikingly for two-meter temperature, precipitation and total cloud cover, three important quantities where both the mean bias and the MSE were reduced. The distribution of runoff, a key driver of land-atmosphere interaction in EC-Earth, was also improved by all three schemes. This demonstrates that the inclusion of stochastic schemes can have a beneficial impact on a models long-term climate mean state.

On the other hand, none of the schemes are able to improve the representation of the QBO, and the Hadley cell becomes too strong and widens too far polewards with ISPPT and LAND. This latter point is likely related to changes in evaporation: while the spatial changes are targeting model biases, leading to a reduced MSE, the overall amplitude of the change is too large. The impact of this is seen most clearly in the energy budget, where both schemes significantly reduced the net surface energy from the relatively realistic levels attained in the deterministic model. However, it is critical to recall that these schemes

have not been tuned, while the deterministic version of EC-Earth used as a reference has been extensively tuned, specifically when it comes to having a realistic energy budget. Tuning parameters for EC-Earth include constants that regulate processes such as entrainment and convection in the atmospheric case, and runoff in the LAND scheme case. In particular, the intensity and frequency of convection/runoff is modulated by these parameters, and tuned in a way to achieve realistic mean states. Non-linear impacts of stochastic schemes can lead to strong impacts on both the intensity and frequency of these processes, as

our experiments have shown, and this significantly alters the mean state of the model.

It is therefore clear that the inclusion of a stochastic scheme must be treated in the same way as the inclusion of any other new parametrisation scheme, in that it will typically require a full re-calibration of the model parameters. By doing so, one may be able to obtain all the benefits to second-order diagnostics in the climate model in question (ENSO, the Asian monsoon, the MJO, European blocking, etc.) while still maintaining a realistic mean state and energy budget. In fact, given the improvements

seen in key regional biases in our experiments, such a tuning procedure could potentially lead to a notably improved mean-state compared to a deterministic model. This will be examined further as part of the PRIMAVERA project, where these schemes will be tested in a fully coupled atmosphere-ocean framework. This work, presented in a future paper, will also include the stochastic ocean and sea-ice schemes and thereby examine the impact of adding stochasticity in every component.




*Code availability.* The EC-Earth source code is available on an SVN repository, and can be obtained by requesting permission via the EC-Earth website http://www.ec-earth.org. The source code for all the stochastic schemes developed and used in this paper can be found in branch r4145-stochastic-ecearth in said repository. The branch of EC-Earth used to generate the simulations considered is titled r3345-oxtest1-isppt.

*Author contributions.* K.S. carried and processed the EC-Earth simulations, analysed output, produced all figures except figure 9, wrote all
sections except sections 2.2, 2.3 and 7.1, and prepared the manuscript for publication. H.M.C. developed and did initial testing of the ISPPT scheme, wrote section 2.2 and 7.1 and produced figure 9. D.M. developed and did initial testing of the stochastic land scheme, wrote section 2.3 and contributed to analysis of the impact on the hydrological budgets. S.J. contributed to analysis on the mean-state impact (sections 4.1, 5.1 and 6.1). T.N.P. secured the Horizon 2020 grant money instrumental for the undertaking of this project, and provided strategical oversight.

*Competing interests.* We declare there are no competing interests present.

*Acknowledgements.* K.S. and T.N.P. acknowledge funding from the European Commission under Grant Agreement 641727 of the Horizon 2020 research programme. H.M.C. acknowledges funding from the Natural Environment Research Council under grant number NE/P018238/1. D.M. acknowledges funding from the EU-FP7 project SPECS (grant agreement 308378). S.J. is contributing to the project M3 of the Collaborative Research Centre TRR 181 "Energy Transfer in Atmosphere and Ocean" funded by the Deutsche Forschungsgemeinschaft (DFG,
German Research Foundation) – Project number 274762653. We especially thank the Climate Prediction group at the Barcelona Supercomputing Center for providing initial condition files for our experiments.



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





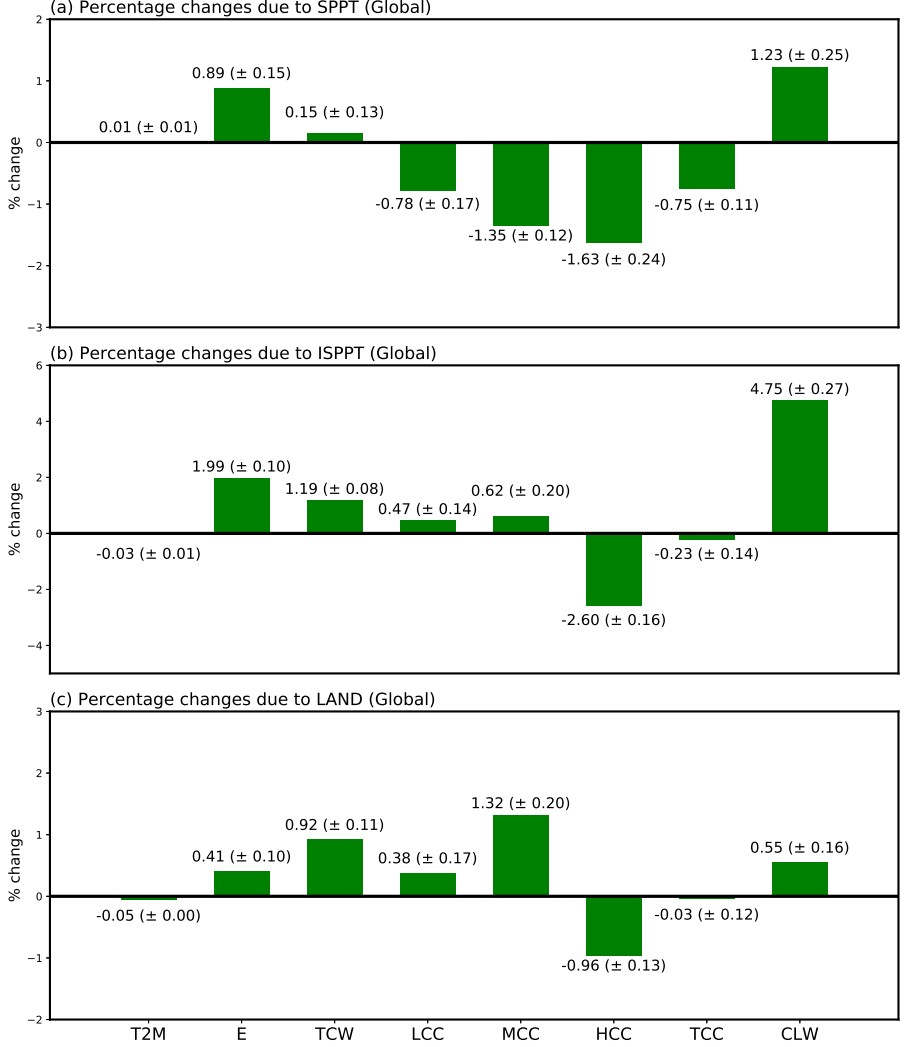

**Figure 1.** Average percentage change, relative to CTRL, in global mean of (a) SPPT simulations, (b) ISPPT and (c) LAND. Variables shown are, in order, two-meter temperature (T2M), evaporation (E), total column water (TCW), low cloud cover (LCC), mid-level cloud cover (MCC), high-level cloud cover (HCC), total cloud cover (TCC) and cloud liquid water (CLW). Uncertainty estimates are twice the standard deviation of the five individual differences. Note: due to conservation of water, the change in precipitation (not shown) is almost exactly equal to the change in evaporation.





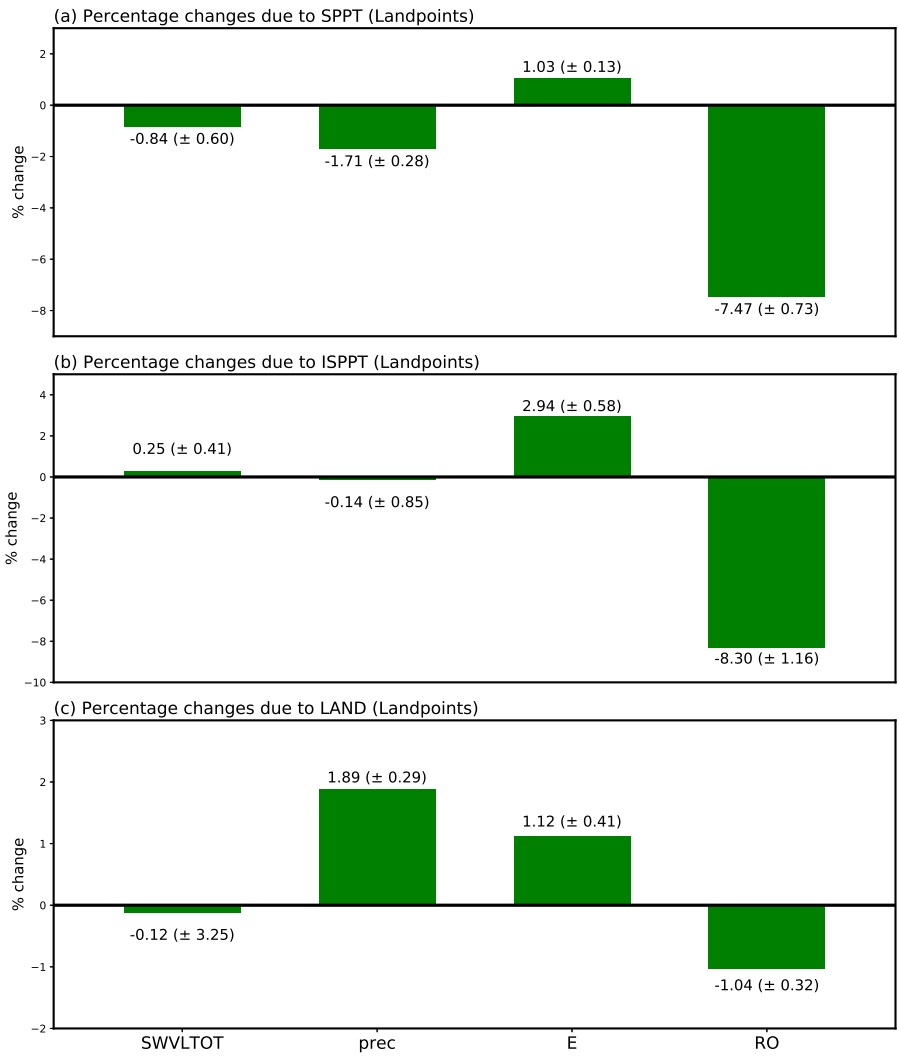

**Figure 2.** Average percentage change, relative to CTRL, in global mean of (a) SPPT simulations, (b) ISPPT and (c) LAND. All data was restricted to land-points only. For LAND, Antarctica was also excluded. Variables shown are, in order, total soil water content (SWVLTOT), precipitation (prec), evaporation (E) and runoff (RO). Uncertainty estimates are twice the standard deviation of the five individual differences.





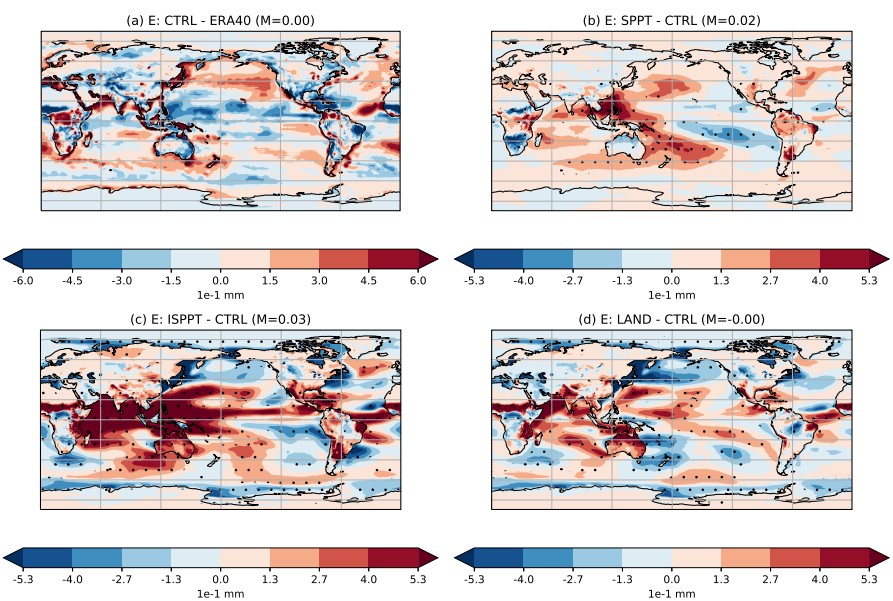

**Figure 3.** Mean differences in evaporation between (a) CTRL and ERA40, (b) SPPT and CTRL, (c) ISPPT and CTRL and (d) LAND and CTRL. Stipling indicates regions where the difference is statistically significant to the 95% confidence interval, as measured by a two-tailed T-test. Note the difference in scales between (a) and (b)-(c)-(d).





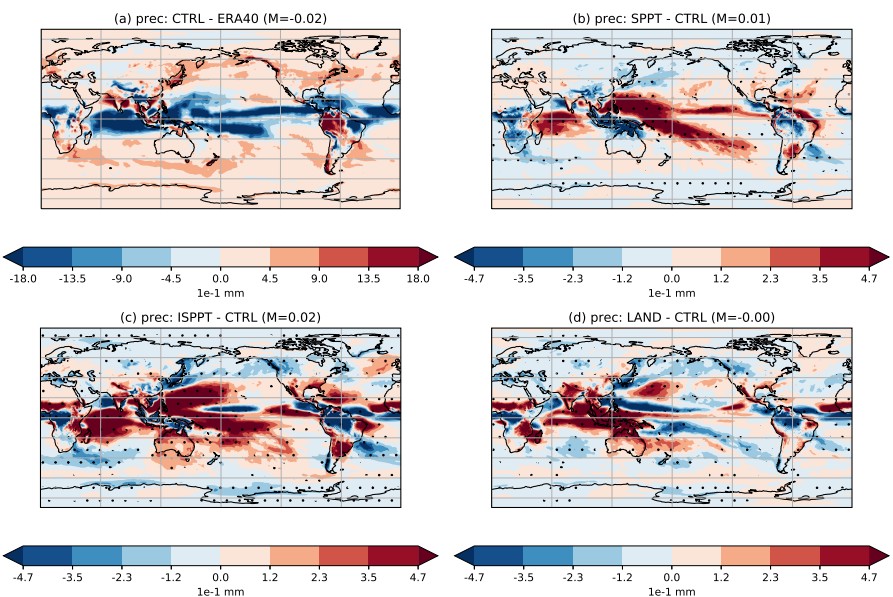

**Figure 4.** Mean differences in precipitation between (a) CTRL and ERA40, (b) SPPT and CTRL, (c) ISPPT and CTRL and (d) LAND and CTRL. Stipling indicates regions where the difference is statistically significant to the 95% confidence interval, as measured by a two-tailed T-test. Note the difference in scales between (a) and (b)-(c)-(d).



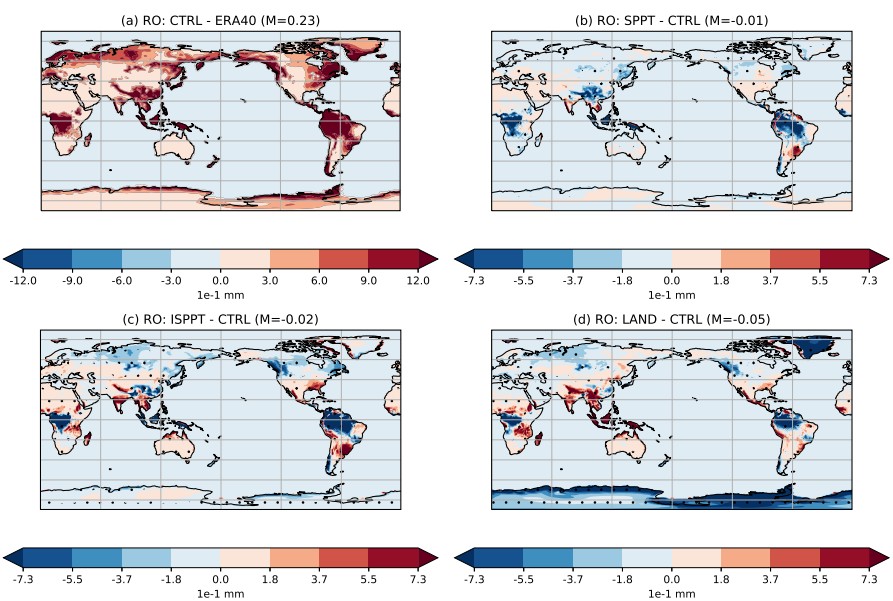

**Figure 5.** Mean differences in runoff between (a) CTRL and ERA40, (b) SPPT and CTRL, (c) ISPPT and CTRL and (d) LAND and CTRL. Stipling indicates regions where the difference is statistically significant to the 95% confidence interval, as measured by a two-tailed T-test. Note the difference in scales between (a) and (b)-(c)-(d).





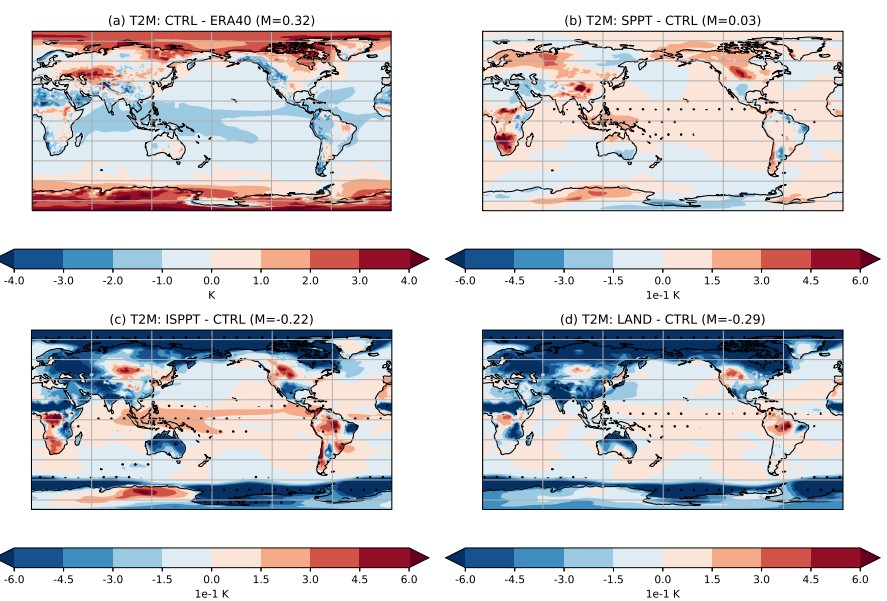

**Figure 6.** Mean differences in two-meter temperature between (a) CTRL and ERA40, (b) SPPT and CTRL, (c) ISPPT and CTRL and (d) LAND and CTRL. Stipling indicates regions where the difference is statistically significant to the 95% confidence interval, as measured by a two-tailed T-test. Note the difference in scales between (a) and (b)-(c)-(d).





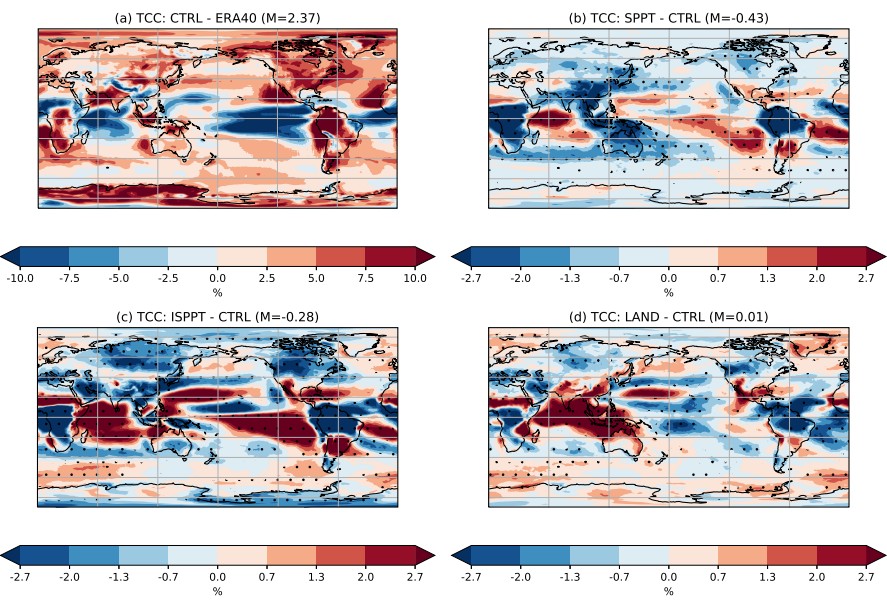

**Figure 7.** Mean differences in total cloud cover between (a) CTRL and ERA40, (b) SPPT and CTRL, (c) ISPPT and CTRL and (d) LAND and CTRL. Stipling indicates regions where the difference is statistically significant to the 95% confidence interval, as measured by a two-tailed T-test. Note the difference in scales between (a) and (b)-(c)-(d). The units of % indicate the proportion of the column occupied by clouds.





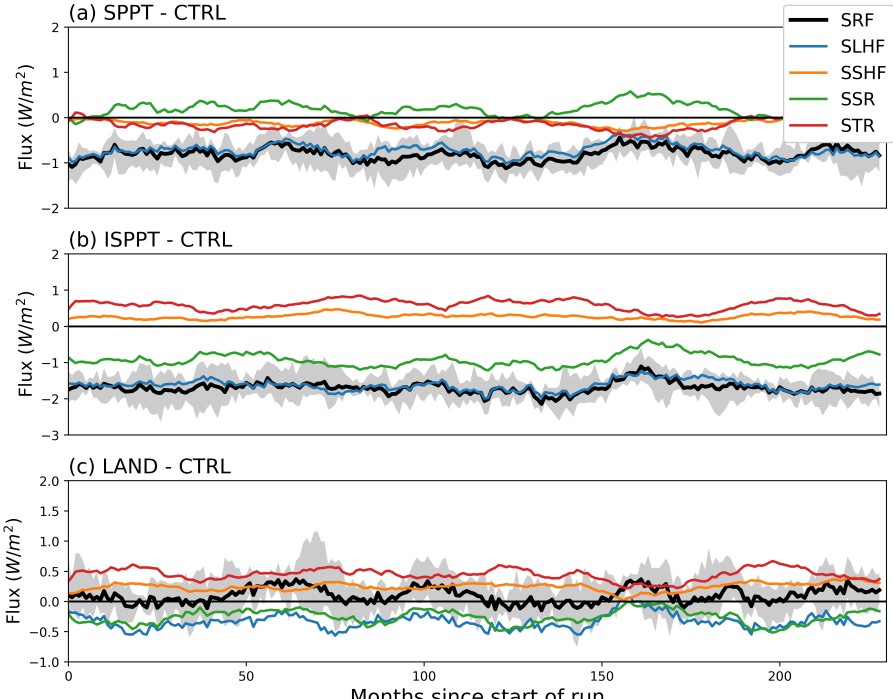

**Figure 8.** Global mean time series of energy fluxes for (a) SPPT minus CTRL, (b) ISPPT minus CTRL, and (c) LAND minus CTRL. Fluxes shown are latent heat flux (SLHF, blue), sensible heat flux (SSHF, green), surface thermal radiation (STR, mauve), surface solar radation (SSR, red) and net surface energy (SRF, black). Note the IFS convention that downward fluxes are positive and upward fluxes (such as sensible and latent heat flux) are negative. SRF is the sum of the other fluxes. The black shading captures two standard deviations around the SRF mean as sampled from the five individual differences. Timeseries have been smoothed with a 12-month running mean to remove the seasonal cycle.



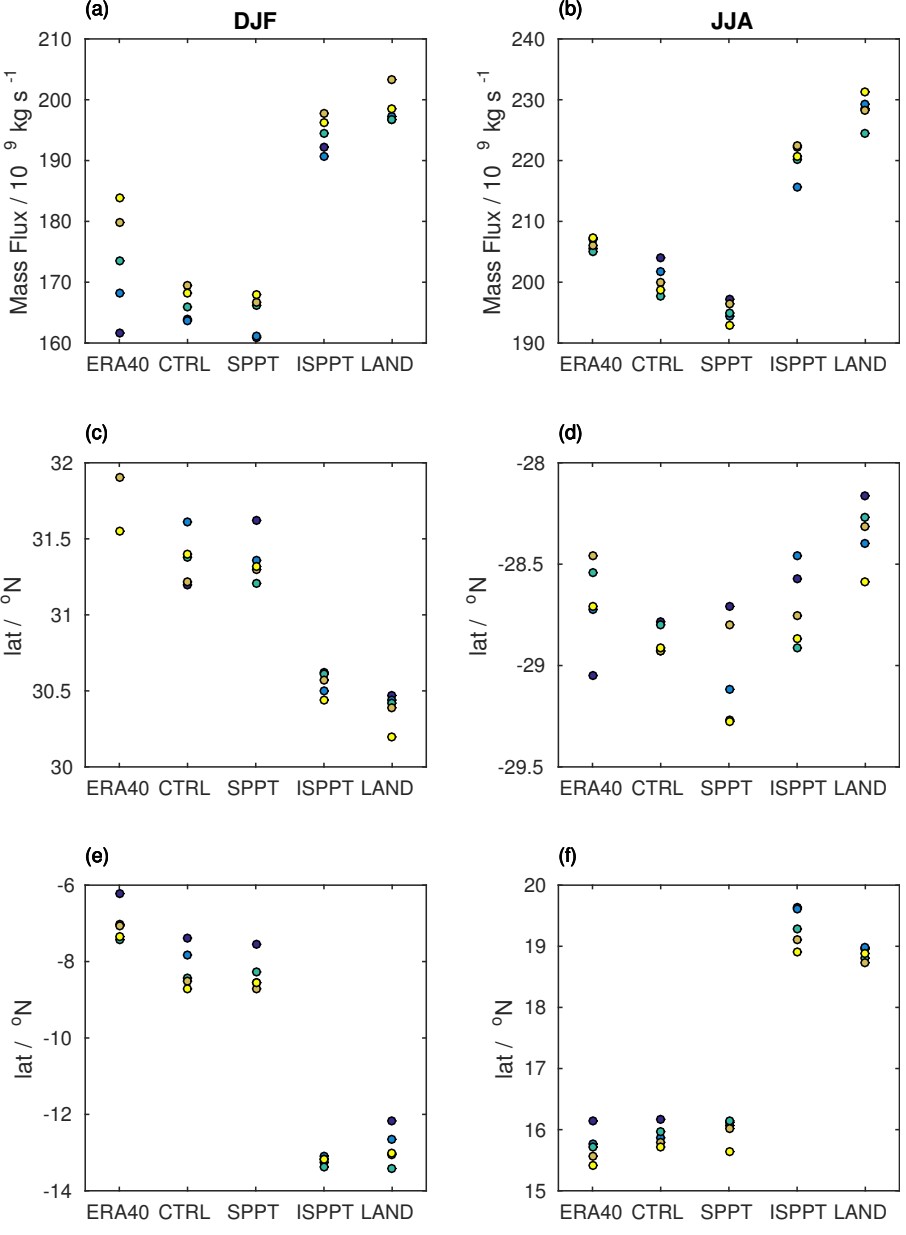

**Figure 9.** Impact of stochastic parametrisations on the Hadley circulation. The magnitude of the overturning circulation indicated by the maximum of the streamfunction of the dominant cell in (a) DJF and (b) JJA for each scheme. The latitude of the downwelling branch of the dominant (winter hemisphere) cell in (c) DJF and (d) JJA. The latitude of the upwelling branch of the dominant (winter hemisphere) cell in (e) DJF and (f) JJA. The data diagnostic is shown for each of the five AMIP simulations in turn, the darker colours indicate earlier years, ranging from dark blue (1960–1980) to yellow (1980–2000).





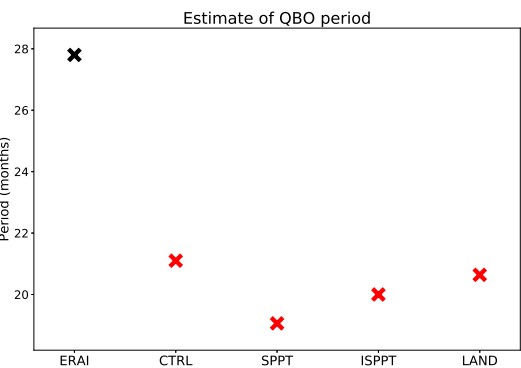

**Figure 10.** Estimate of the QBO period, as measured using equatorial zonal winds at 50hPa. The reanalysis product ERA-Interim is shown for comparison.





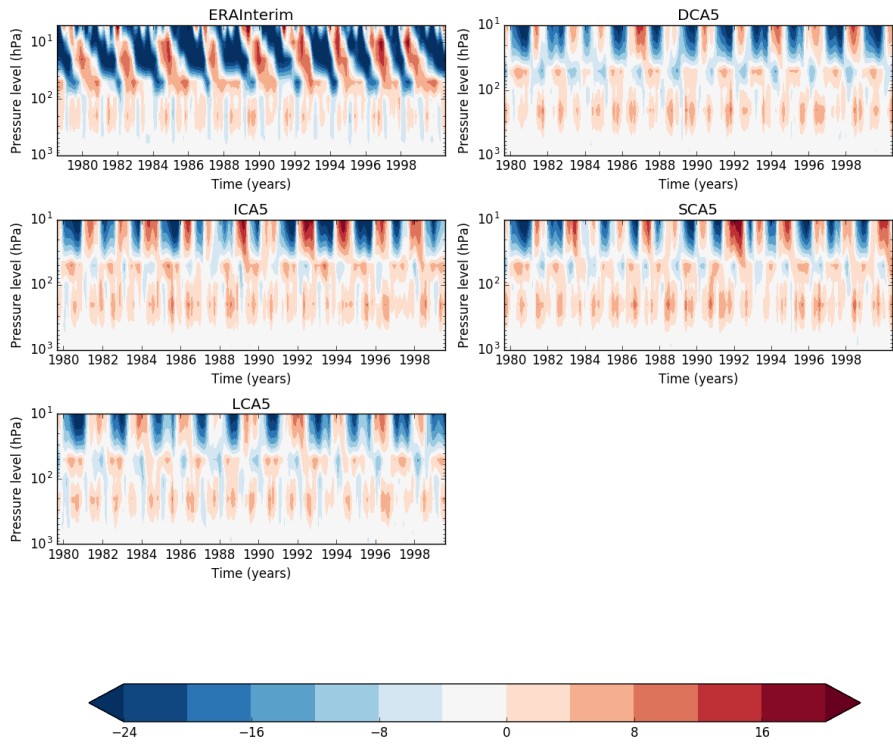

**Figure 11.** Visualisation of the QBO (time-pressure level section of zonal winds between 10S and 10N with the seasonal mean removed) for the ensemble member covering 1980-2000, the period overlapping the most with ERA-Interim. Pressure levels are plotted according to a logarithmic scale for ease of interpretation. Here DCA5 is the CTRL simulation, ICA5 the simulation with ISPPT, SCA5 the one with SPPT, and LCA5 the one with LAND scheme. Units are $ms^{-1}$.




**Table 1.** Globally averaged surface energy fluxes for the CTRL simulation. The values in the Observed row are the estimates from (Trenberth et al., 2009) for the period 2000-2004. STR = net surface thermal radiation, SSR = net surface solar radiation, SLHF = surface latent heat flux, SSHF = surface sensible heat flux, SRF = net surface energy. Units are $W/m^2$ in all cases.

|           | STR  | SSR   | SLHF | SSHF | SRF |
|-----------|------|-------|------|------|-----|
| 1991-1995 | 61.2 | 163.8 | 82.7 | 19.3 | 0.6 |
| 1991-2000 | 61.1 | 164.2 | 82.8 | 19.4 | 1.0 |
| Observed  | 63   | 161   | 80   | 17   | 0.9 |





**Table 2.** Mean-square errors between the spatial means of reanalysis (ERA40) and each EC-Earth configuration (CTRL, SPPT, ISPPT and LAND) for T2M (two-meter temperature), Prec (precipitation), E (evaporation), TCC (total cloud cover), RO (total runoff), TCW (total column water) and SWS (wind-speeds at 1000hPa). Values where a stochastic scheme significantly reduced the error (see section 3.2) have been bolded.

|      | CTRL   | SPPT       | ISPPT      | LAND       |
|------|--------|------------|------------|------------|
| T2M  | 2.545  | 2.551      | **1.940**  | **1.851**  |
| Prec | 0.809  | **0.754**  | **0.724**  | **0.764**  |
| E    | 0.071  | 0.075      | **0.065**  | **0.059**  |
| TCC  | 0.0035 | **0.0030** | **0.0028** | 0.0035     |
| RO   | 0.457  | **0.393**  | **0.366**  | **0.405**  |
| TCW  | 1.454  | 1.471      | 1.546      | 1.557      |
| SWS  | 0.445  | 0.463      | 0.479      | 0.448      |