# Peer review of "Progress Towards a Probabilistic Earth System Model: Examining The Impact of Stochasticity in the Atmosphere and Land component of EC-Earth v3.2"

_Geoscientific Model Development, 2018_

## Referee Comment (RC1) · Anonymous Referee #1 · 17 Mar 2019

Introducing the probabilistic Earth-System Model: Examining the impact of stochasticity in EC-Earth v3.2 By Kristian Strommen, Hanna Christensen, Dave MacLeod, Stepahn Juricke and Tim Palmer.

General Comment: This paper compares different stochastic perturbation schemes for the parametrization tendencies in the context of the EC-Earth model. STTP, ISTTP and an stochastic parameter perturbation for soil model are compared. The impact of stochastic parametrizations upon atmospheric and coupled models is a highly relevant topic and this paper performs experiments towards the implementation of stochastic-coupled modeling systems. The paper shows the impact of these stochastic

parametrization approaches upon a global model and particularly over the mean state over a relatively long simulation.

The paper needs to be improved in some important aspects before it can be published.

Major points

1) The order of the figures needs to be revised. (E.g. line 17, page 7, Figure 3 is mentioned before Figure 2).

2) Some choices in the implementation of the stochastic perturbation needs to be motivated. For example what is the motivation behind the 3 temporal and spatial scales associated to STTP and ISTTP (Line 8, page 4). Also why the amplitude of the multiplicative perturbation factor is tappered in the boundary layer (given that the PBL scheme is a source of the kind of model errors that the stochastic perturbations are trying to represent?). Another choice that is not motivated is the use of parameter perturbation instead of stochastic tendency perturbation in the LAND experiment (Section 2.3).

3) What is the motivation behind using the same perturbation for convection and large scale condensation in the ISPPT approach? The tendencies produced by these parametrizations are sometimes anti-correlated since when convection fails to remove instability from an atmospheric column large scale condensation tries to do that.

4) It would be good to provide more discussion about the pathways in which the stochastic perturbations can change the mean. I agree in that the impact of SPPT and ISPPT suggests that the convective scheme is activated more frequently, however the discussion on how the stochastic perturbations can lead to this is not clear (e.g. line 14, page 10). In the discussion section it is stated that some perturbations can trigger convection in areas in which the unperturbed state has conditions close to those required to activate the convective scheme. However the opposite is also possible, some columns in which the unperturbed state is sufficient for the initiation of deep moist convection can be perturbed leading to a state in which these conditions are not met any more.

5) As part of a first evaluation of the impact of SPPT, ISPPT and LAND upon the EC-Earth model it would be good to present some scores related to atmospheric circulation. Like for example MSE and biases for wind at different atmospheric levels and also for temperature at these same levels. The goal of the paper is focused on surface fluxes, but atmospheric circulation is also examined by studying for example the impact upon the Hadley cell. Although the impact upon the Hadley cell is relevant (particularly because SPPT and ISPPT seems to produce a large impact upon tropical convection), it would be good to provide these other scores for comparison with other systems.

6) It is not clear for me what is the motivation to study the QBO in the context of this paper. I understand that the impact upon different aspects of the atmospheric dynamics should be investigated but the inclusion of this particular aspect in a first evaluation has to be better motivated.

Minor points

1) Line 10, page 5. What does exactly mean that parameters are correlated? Estimated parameters based on observation studies show that the value of these parameters in different soil types and conditions are correlated or that the joint sensitivity of these two parameters shows a certain degree of compensation between the impact of these two parameters (i.e. the effect of the increase in one of the parameters can be compensated by changes in the other parameter).

2) Why performing 5 periods of 20 years each instead of a longer simulation. Using 5 different periods as ensemble members can artificially increase the ensemble spread and reduce the significance of the results. Also spin-up issues may be more important when several shorter periods are considered, particularly in the soil variables.

3) Line 29, page 4. Remove parenthesis and "for details".

4) Line 29, page 6. Missing space before Spatial. Also indicate instead of indiate.

5) Since convective precipitation is part of the products generated by the convective scheme and is linked to the other tendencies, is precipitation rate perturbed in the same way as the other tendencies produced by the parametrization? Same question but for the large scale condensation scheme.

6) Line 6 page 14. More frequent convective scheme activation can also explain why the PBL is drier.

7) Figure 4 a, shows the biases in the precipitation for the control run. This bias pattern is strong and shows a clear maximum in the tropics. The authors indicate that the control configuration has been extensively tuned, however has the tunning been performed with this same model resolution?

8) Line 16, page 15. Changes in the Hadley cell are caused by changes in evaporation? Or these two changes are driven by changes in tropical convection?

9) Figure 6: The changes in T2m over the sea ice in the ISPPT and LAND are very strong. It is surprising to see these changes in both experiments since none of these experiments seems to directly affect the sea-ice parametrization in any way (SPPT for example do not show a strong change in bias in this region). I suggest to check the sea-ice distribution and temperature in these experiments.

10) Figure 11. I suggest to use the same names as in the rest of the manuscript.

11) Figure 8. Please correct the caption since the colors do not correspond to the ones on the legend (I assumed that the legend is correct).

12) Line 10, page 8. This sentence is not clear, I can not see "each model simulation" but something that seems to be the mean of all simulations.

13) It would be better to use the same color scale for all panels in figures 3, 4, 5, 6 and 7. In most cases the range is similar. Another possibility is to show in all cases the

bias with respect to ERA (again since the magnitudes are similar this should clearly show the improvement produced by the stochastic schemes and would be more easy to analyze). Also in this figures indicate what "M=" stands for. I assumed that this is the mean bias over the global domain.

14) Since the main goal is to perform analysis towards the development of a coupled stochastic modeling system, why a SPPT+LAND or ISPPT+LAND experiments where not performed?

---

## Referee Comment (RC2) · Anonymous Referee #2 · 3 May 2019

General comments ================

The authors present a well-written manuscript about the introduction of stochastic schemes into the atmosphere and land-surface scheme of EC-Earth, testing the impact on the various schemes on the mean biases as well as on variability. Stochasticity in a climate or earth-system model context is still rather novel and the work presented here is a valuable addition in the field. I therefore recommend the manuscript for publication in GMD after some necessary revision.

Specific comments ================

Title: the title mentions the "Probabilistic Earth-System Model" while indeed all the

work has been done entirely with the atmosphere component of EC-Earth only. In my eyes the title this suggests more than what the manuscript delivers and to avoid any too far reaching expectation I'd therefore suggest to modify the title to something more adequate.

Section 2.1, 1st paragraph: there is no need to go into details about EC-Earth's ocean component or coupler because they are irrelevant in this work.

Section 2.2, eq 1: what is "the i'th physics parameterization scheme"? You mention the schemes explicitly towards the end of Sec 3.1, why don't you list them here already?

Section 2.2, eq 1: are the tendencies for all variables perturbed by the same r? Or are there different perturbations for the different variables?

Section 2.2 , eq 1: are the pertubations constant in time, or do they vary from one timestep to the next?

Section 2.2, end of 1st paragraph: you say "perturbation is limited between 0 and 2" but shouldn't that be between -1 and 1? Even with r=-1 we still get that P_hat has the same sign as P, or?

Section 3.1, p.6 l.13: why do you use ERA-40 or ERA interim for the evaluation? These re-analysis datasets belong to the same model "family" as EC-Earth and may therefore share common biases, in particular where the re-analysis are largely a model product not constrained by observations. I would prefer the evaluation to be done against a different re-analysis dataset, but that may be too much to ask for at this stage. In any case, it would be good if the similarity between EC-Earth and the model used for the re-analysis would be clearly stated.

Section 3.4: how good is it to compare only the last 10 (5) years from the last ensemble member against observations? Is that short time period really sufficient to confirm or reject the fitness of EC-Earth? I don't understand why you don't average 10-20 year from all ensemble members (start dates) and compare against the corresponding time

period of the re-analysis (see also comment above about choice of re-analysis data).

Section 4,5,6: when you talk about cloud water do you mean the gridpoint value of the in-cloud value? I would suspect it's the gridpoint value, and if that's the case then the reduction in cloud cover would imply an even larger increase in the in-cloud water content with all the consequences for optical thickness and cloud microphysics. Good if you could clarify on this point.

Section 4.2,5.2,6.2 and Fig 8: I don't see a point in making a timeseries of the energy budget, the interesting aspect for a climate model is how well it simulates the average flux and its variability compared to observations (re-analysis). For that reason it would make more sense to average the biases in the fluxes and present them as a table or barplot similar to how you did in Figs 1 and2.

Section 7: what is the motivation for selecting the Hadley circulation and the QBO as test cases for the stochasticity? Why not NAO or PNA as other prominent modes of atmospheric variability?

Section 7.1 Fig 9: It is not easy to easy to distinguish the different colors of the dots (I am slightly color blind) and you should consider presenting the data in a different way, e.g. by using different symbols for the different members/time periods.

Section 7.2: what is the reason for evaluating QBO only for the last ensemble member? Why not using the results from all start dates?

Section 8, 1st sentence: you cannot blame the absence of a more process-oriented analysis on the lack of available data because you have designed the experiment and its output.

Section 8.1, p.14 l.7: it's not clear to me how an increased in-cloud liquid water content could steepen the near-surface humidity gradient. Could you explain better what you mean?

Section 8.2, p.15 l.13: what do you mean with runoff being a key driver of landatmosphere interaction? Isn't runoff simply the difference between P-E and the amount of water absorbed by soil? It's a residual, not a driver, or?

Section 8.2, p.15 l.22: why do you call runoff a tuning parameter in the LAND case? Runoff is an important diagnostics of the model that can be used to tune the model, but runoff itself isn't a tuning parameter.

Code availability: EC-Earth is licensed and not openly available, it's not sufficient for any presumptive user to request access. I would suggest you check the guidelines of GMD that regulate code availability and re-phrase this section. (https://www.geoscientific-model-development.net/about/code_and_data_policy.html)

Technical corrections =====================

Sorry, I am not a native English speaker and cannot really comment on the language, but two things struck my eye:

p.2 l.12: "However" seems to be inappropriate in this sentence.

p.15 l.15: shouldn't it rather be "...none of the schemes _is_ able..."?
* * *

---

## Author Comment (AC1) · 20 May 2019

**RESPONSE TO REVIEW #1 OF GMD MANUSCRIPT:**

**"Introducing the Probabilistic Earth-System Model: Examining The Impact of Stochasticity in EC-Earth v3.2"**

We thank the reviewer (hereby referred to as Reviewer #1) for their comments, which have helped us improve the clarity and quality of our manuscript.

**MAJOR POINTS**

1) **Reviewer #1:** *The order of the figures needs to be revised. (E.g. line 17, page 7, Figure 3 is mentioned before Figure 2).*

**Our response:** Yes, this was a clear oversight on our part. The revised manuscript now has figures numbered in the order they appear.

2) **Reviewer #1: "***Some choices in the implementation of the stochastic perturbation needs to be motivated. For example what is the motivation behind the 3 temporal and spatial scales associated to STTP and ISTTP (Line 8, page 4). Also why the amplitude of the multiplicative perturbation factor is tappered in the boundary layer (given that the PBL scheme is a source of the kind of model errors that the stochastic perturbations are trying to represent?). Another choice that is not motivated is the use of parameter perturbation instead of stochastic tendency perturbation in the LAND experiment (Section 2.3).***"*

**Our response:** We initially omitted some of these details for the sake of brevity, as most of the answers are covered in the cited papers. In response to this comment, we have now addressed these questions in the revised manuscript. The answers are as follows.

The three temporal and spatial scales associated to the SPPT/ISPPT perturbations come from the assumption that model error can come both from processes that happen at very fast timescales (and, therefore, small spatial scales), and also, independently, for processes happening on very slow timescales (and, therefore, large spatial scales). The three patterns are a compromise aimed at capturing a plausible range of relevant atmospheric processes.

The tapering in the boundary layer was implemented due to model stability issues. Numerical instabilities begin to accumulate when perturbations are added in the lowest model levels due to the delicate balance between model dynamics and vertical transport in this region.

Finally, for the stochastic land scheme, in fact tendency perturbations had been tested initially, but were found to have little to no impact on soil moisture in the model.

Parameter perturbations were therefore done instead, as these produce changes in soil moisture that more plausibly capture the model uncertainties.

3) **Reviewer #1: "***What is the motivation behind using the same perturbation for convection and large scale condensation in the ISPPT approach? The tendencies produced by these parametrizations are sometimes anti-correlated since when convection fails to remove instability from an atmospheric column large scale condensation tries to do that.***"**

**Our response:** In the IFS, which forms the atmospheric component of EC-Earth, moist processes are split up into convection and large-scale condensation. The convection scheme transports moisture aloft, where it is detrained. This forms an input to the large-scale condensation scheme, which determines whether supersaturation has been reached and therefore how much cloud should be present. If needed, it then removes water vapour from the atmosphere with a corresponding increase in liquid water content. The intimate relationship between convection and large-scale condensation, whereby the outputs from one form the inputs to the other, motivated us to perturb these schemes together. Tests in an NWP setting indicate that grouping moist processes in this way gave skilful weather forecasts (Christensen et al, 2017, QJRMetS).

4) **Reviewer #1: "***It would be good to provide more discussion about the pathways in which the stochastic perturbations can change the mean. I agree in that the impact of SPPT and ISPPT suggests that the convective scheme is activated more frequently, however the discussion on how the stochastic perturbations can lead to this is not clear (e.g. line 14, page 10). In the discussion section it is stated that some perturbations can trigger convection in areas in which the unperturbed state has conditions close to those required to activate the convective scheme. However the opposite is also possible, some columns in which the unperturbed state is sufficient for the initiation of deep moist convection can be perturbed leading to a state in which these conditions are not met any more.***"**

**Our response:** We would like to clarify that we do not assert that SPPT/ISPPT necessarily would be expected to trigger convection more or less frequently. Rather, our main assertion is that the inherent asymmetry in condensation triggers means that even a totally symmetric perturbation could lead to an increase in the mean cloud water content. This is because, while a perturbation away from e.g. deep moist convection would not be expected to change the cloud water content in the column (for that timestep), a perturbation towards it may trigger water to condense, which increased cloud water content. In other words, the impact of symmetric perturbations would be expected, if anything, to increase cloud water.

This increase in cloud water is suggested as a leading order mechanism (or pathway) towards larger scale changes with SPPT/ISPPT. This hypothesis is explained in the original manuscript on page 14, lines 9-16 (see the whole paragraph for a full discussion).

The comment the reviewer refers to (line 14, page 10) is specifically about precipitation *extremes*, which had been observed to increase with SPPT in previous studies. This was suggested as a potentially important pathway towards the observed changes in soil moisture.

The revised manuscript now contains some clarifying remarks to this effect in both sections.

5) **Reviewer #1:** *"As part of a first evaluation of the impact of SPPT, ISPPT and LAND upon the EC-Earth model it would be good to present some scores related to atmospheric circulation. Like for example MSE and biases for wind at different atmospheric levels and also for temperature at these same levels. The goal of the paper is focused on surface fluxes, but atmospheric circulation is also examined by studying for example the impact upon the Hadley cell. Although the impact upon the Hadley cell is relevant (particularly because SPPT and ISPPT seems to produce a large impact upon tropical convection), it would be good to provide these other scores for comparison with other systems."*

**Our response:** We thank the reviewer for the suggestion, which we agree would be a helpful way to allow for easy comparison with other studies. We have extended Table 2 to add the MSE of temperature and zonal wind fields at various levels. This also helps illuminate the reviewer's next question, and our answer to it.

6) **Reviewer #1:** *"It is not clear for me what is the motivation to study the QBO in the context of this paper. I understand that the impact upon different aspects of the atmospheric dynamics should be investigated but the inclusion of this particular aspect in a first evaluation has to be better motivated."*

**Our response:** The reviewer makes a good point: this was not adequately motivated. Our primary motivation here comes from the paper Leutbecher et al. (2017), *Stochastic representations of model uncertainties at ECMWF: State of the art and future vision,* where the impact of SPPT on short to medium range forecasts is considered. On page 11 of this paper, it is documented that the biggest degradation of SPPT on the version of the IFS considered is on the upper level winds, where the QBO dominates. The MSE computations we added in Table 2 (c.f. our response to the previous comment) show similar behaviour. This is raised in ibid as being a point of concern due to the growing body of literature suggesting that the QBO is an important driver of European climate at seasonal timescales. We therefore wished to identify if a similar degradation occurs in the EC-Earth model for the schemes considered.

We have now added this motivation to the introduction of section 7.

**MINOR POINTS**

1) **Reviewer #1:** *"Line 10, page 5. What does exactly mean that parameters are correlated? Estimated parameters based on observation studies show that the value of these parameters in different soil types and conditions are correlated or that the joint sensitivity of these two parameters shows a certain degree of compensation between the impact of these two parameters (i.e. the effect of the increase in one of the parameters can be compensated by changes in the other parameter)."*

**Our response:** We meant the former. Estimations of parameters based on observation studies indicate that the two parameters show correlation across soil types. This correlation is neither zero nor one, indicating that the parameters are related whilst showing some independence. We therefore chose not to perturb the two parameters with the same pattern, nor with two entirely independent patterns. Rather, we introduce some dependence between the perturbations through definition of a third pattern, as described in the text.

The rationale for this is that the observed correlation suggests that some parts of parameter space may be unphysical (for instance extremely high values of gamma and low values of alpha). An independent perturbation would be likely to access these regions of parameter space (for instance simultaneous values of 1+r=1.8 for gamma and  1+r=0.1 for alpha). By tethering both parameters to a third base pattern, this possibility is reduced and the parameters are perturbed in a more similar way, whilst retaining some independence in the perturbation.

We have slightly rephrased the description in this section to make it clear that the correlation is based on observational studies.

2) **Reviewer #1:** *"Why performing 5 periods of 20 years each instead of a longer simulation. Using 5 different periods as ensemble members can artificially increase the ensemble spread and reduce the significance of the results. Also spin-up issues may be more important when several shorter periods are considered, particularly in the soil variables."*

**Our response:** When constructing the experimental protocol, the authors recognized that there were several aspects it would be good to test, but that there were insufficient computer units to run all the configurations desired. We decided that the most important goal was to produce runs where the significance of any impacts could best be detected: for this reason, it was decided to produce an ensemble of simulations for each scheme as opposed to a single long run. The authors believe that comparison of distinct ensemble members allows for the most transparent and accurate assessment of the uncertainty in the computed metrics. While such an assessment from a single longer run would be possible by using techniques such as subsetting the data and/or bootstrap resampling, these techniques are essentially just trying to artificially create distinct ensemble members within the longer timeseries anyway. Therefore, simply producing an actual ensemble from definitely distinct initial conditions gives the cleanest methodology, in our opinion. We are not aware of any

reason why this may increase the spread in a way which is excessive compared to that diagnosed with other experimental protocols.

The initialisation of each ensemble member to a slightly different time period also allows us to cleanly assess any dependence of the rapid response on the initial ocean and land state. Spin up issues may have been expected to be problematic for the land state, but in practice we found that the main changes were the same across all the time periods, implying that the impact of the schemes are quite rapid and do not require a longer period to assess accurately.

We have included some more discussion on why our experimental protocol was chosen, based on the above discussion, in the revised version of the paper and hope that this will satisfy the reviewer.

3) **Reviewer #1:** *"Line 29, page 4. Remove parenthesis and "for details"."*

**Our response:** We made the suggested change.

4) **Reviewer #1:** *"Line 29, page 6. Missing space before Spatial. Also indicate instead of indiate."*

**Our response:** We made the correction.

5) **Reviewer #1:** *"Since convective precipitation is part of the products generated by the convective scheme and is linked to the other tendencies, is precipitation rate perturbed in the same way as the other tendencies produced by the parametrization? Same question but for the large scale condensation scheme."*

**Our response:** Thank you for your insightful question. You are right, for complete consistency the fluxes in precipitation and evaporation should be perturbed using the same pattern as for SPPT. However, this is not currently implemented in SPPT. This could be part of the reason why the scheme does not conserve water, which is why the `humidity fix' has been implemented to correct this. Having said that, testing is underway to develop a more consistent approach whereby precipitation and evaporation are perturbed, to bypass the need for the `humidity fix'.

6) **Reviewer #1:** *"Line 6 page 14. More frequent convective scheme activation can also explain why the PBL is drier."*

**Our response:** We thank the reviewer for the insightful comment: we have added a comment on this in this section of the revised manuscript.

**7) Reviewer #1:** *"Figure 4 a, shows the biases in the precipitation for the control run. This bias pattern is strong and shows a clear maximum in the tropics. The authors indicate that the control configuration has been extensively tuned, however has the tunning been performed with this same model resolution?"*

**Our response:** The model was indeed tuned at the T255 spectral resolution which we used in this study. The tuning procedure for EC-Earth is carried out in order to obtain a realistic energy budget, with a particular focus on the net surface energy flux. In particular, precipitation biases are not directly tuned. Therefore, while the deterministic model has little bias in the energy fluxes at the surface, it does still have relatively notable biases in key variables like precipitation.

**8) Reviewer #1:** *"Line 16, page 15. Changes in the Hadley cell are caused by changes in evaporation? Or these two changes are driven by changes in tropical convection?"*

**Our response:** The reviewer makes a good point; the manuscript as it stood focused on changes to evaporation (latent heat flux) because of the focus on energy budget changes. However, it is of course possible that the first order impact is on tropical convection, and evaporation changes are a response to this. We have included some discussion of this in section 8.1 (Discussion) and 8.2 (Conclusions).

**9) Reviewer #1:** *"Figure 6: The changes in T2m over the sea ice in the ISPPT and LAND are very strong. It is surprising to see these changes in both experiments since none of these experiments seems to directly affect the sea-ice parametrization in any way (SPPT for example do not show a strong change in bias in this region). I suggest to check the sea-ice distribution and temperature in these experiments."*

**Our response:** Being AMIP style experiments, the sea-ice distribution is a fixed field along with the sea-surface temperatures. Therefore, for both ISPPT and LAND, the cooling seen in the sea-ice regions are necessarily induced via atmospheric circulation changes, which can cause surface temperature (which is dynamic) to change. Discussion of potential mechanisms behind atmospheric circulation changes are included in the manuscript, and have been extended upon based on other comments by both reviewers. We therefore do not expand upon this further.

**10) Reviewer #1:** *"Figure 11. I suggest to use the same names as in the rest of the manuscript."*

**Our response:** We made the suggested change.

**11) Reviewer #1:** *"Figure 8. Please correct the caption since the colors do not correspond to the ones on the legend (I assumed that the legend is correct)."*

**Our response:** We thank the reviewer for pointing out this silly mistake on our part. The caption has been edited to match the legend in the figure.

**12) Reviewer #1:** *"Line 10, page 8. This sentence is not clear, I can not see "each model simulation" but something that seems to be the mean of all simulations"*

**Our response:** Our phrasing was somewhat unclear. We meant `the mean across all the individual differences'. We have rephrased this as follows: `Figure 8(a) shows the mean difference between the five SPPT-simulations and the corresponding CTRL-simulations (i.e. the average across the 5 differences), …'
We hope this rephrasing makes the meaning clearer.

**13) Reviewer #1:** *"It would be better to use the same color scale for all panels in figures 3, 4, 5, 6 and 7. In most cases the range is similar. Another possibility is to show in all cases the bias with respect to ERA (again since the magnitudes are similar this should clearly show the improvement produced by the stochastic schemes and would be more easy to analyze). Also in this figures indicate what "M=" stands for. I assumed that this is the mean bias over the global domain."*

**Our response:** We have added the explanation for "M=" in the captions now, a clear oversight: the reviewer was indeed correct in their assumption of its meaning.
We would like to respectfully disagree with the reviewer on the suggestion to change the spatial maps to show the bias with respect to ERA in each panel, as we believe this would make it harder, not easier, to analyse. As an example, consider the precipitation changes in figure 4. Note that the CTRL bias is nearly 3 times as large in magnitude as the impact of the stochastic schemes (with the former having a peak of ~1.8mm and the latter a peak of ~0.6mm). As a result, showing ERA biases side by side is not very illuminating, as the following figure demonstrates:

[Figure]

Showing the bias relative to CTRL immediately illuminates how the scheme is changing the mean state for all the variables. Colourscales are tailored specifically to each individual variable, and chosen so that the scale encapsulates 2 standard deviations around the mean bias. For a more easily comparable quantitative metric, we have used the mean bias (M=…) and the MSE table. We therefore suggest to leave the figures as they are.

**14) Reviewer #1:** *"Since the main goal is to perform analysis towards the development of a coupled stochastic modeling system, why a SPPT+LAND or ISPPT+LAND experiments where not performed?"*

**Our response:** In fact, such experiments were carried out. These were not ultimately included in the analysis proper, because the impact of SPPT/ISPPT is typically much larger in magnitude than the impact of LAND, which makes it difficult to assess the individual impact of either in a joint experiment. This piece of information should clearly have been included in the paper, and we thank the reviewer for making this point. We have now added a paragraph on this in the Experimental Setup section (section 3.1).

---

## Author Comment (AC2) · 20 May 2019

**RESPONSE TO REVIEW #2 OF GMD MANUSCRIPT:**

**"Introducing the Probabilistic Earth-System Model: Examining The Impact of Stochasticity in EC-Earth v3.2"**

We thank the reviewer (hereby referred to as Reviewer #2) for their comments, which have helped us improve the clarity and quality of our manuscript.

1) **Reviewer #2:** *"Title: the title mentions the "Probabilistic Earth-System Model" while indeed all work has been done entirely with the atmosphere component of EC-Earth only. In my eyes the title this suggests more than what the manuscript delivers and to avoid any too far reaching expectation I'd therefore suggest to modify the title to something more adequate."*

**Our response:** The reviewer makes a fair point. We have now edited the title to "Progress Towards a Probabilistic Earth System Model: Examining The Impact of Stochasticity in EC-Earth v3.2", reflecting the paper as a natural step from the vision outlined in Palmer (2012), "Towards the probabilistic Earth system simulator". This should be more in line with the actual content of the paper.

2) **Reviewer #2:** *"Section 2.1, 1st paragraph: there is no need to go into details about EC-Earth's ocean component or coupler because they are irrelevant in this work."*

**Our response:** We have removed these details.

3) **Reviewer #2:** *"Section 2.2, eq 1: what is "the i'th physics parameterization scheme"? You mention the schemes explicitly towards the end of Sec 3.1, why don't you list them here already?"*

**Our response:** We have now explicitly identified the different physics schemes in section 2.2.

4) **Reviewer #2:** *"Section 2.2, eq 1: are the tendencies for all variables perturbed by the same r? Or are there different perturbations for the different variables?"*

**Our response:** All variables are perturbed in the same way: the tendency vector P has, as its entries, the tendencies for temperature, humidity, and wind fields. It is this vector which is perturbed, implying that all variables are perturbed similarly. We have slightly rephrased the explanation of what P is to emphasize that it contains all these tendencies to clarify this point.

5) **Reviewer #2:** *"Section 2.2 , eq 1: are the pertubations constant in time, or do they vary from one timestep to the next?"*

**Our response:** The perturbations evolve over time as AR(1) processes. We have rephrased the presentation slightly in response to this point, emphasizing both that the perturbation happens at each timestep, and that the perturbation r evolves in time.

6) **Reviewer #2:** *"Section 2.2, end of 1st paragraph: you say "perturbation is limited between 0 and 2" but shouldn't that be between -1 and 1? Even with r=-1 we still get that P_hat has the same sign as P, or?"*

**Our response:** This was incorrectly phrased on our side: the reviewer is entirely correct that r itself is clipped between -1 and 1: this implies that 1+r is clipped between 0 and 2, which is why this range was referenced in the submitted manuscript. We have revised the script accordingly.

7) **Reviewer #2:** *"Section 3.1, p.6 l.13: why do you use ERA-40 or ERA interim for the evaluation? These re-analysis datasets belong to the same model "family" as EC-Earth and may therefore share common biases, in particular where the re-analysis are largely a model product not constrained by observations. I would prefer the evaluation to be done against a different re-analysis dataset, but that may be too much to ask for at this stage. In any case, it would be good if the similarity between EC-Earth and the model used for the re-analysis would be clearly stated."*

**Our response:** The reviewer makes a good point. Experience suggests this is unlikely to be an issue for surface variables such as surface temperature which are strongly constrained by existing observations and therefore do not differ much across reanalysis datasets. However, certain less strongly constrained variables, such as total cloud cover (considered in the paper) may be more affected by this, so we agree it is important to point out.

We have now added a disclaimer addressing this point in section 3.1 covering this point, and we hope the reviewer will find this sufficient.

**8) Reviewer #2:** *"Section 3.4: how good is it to compare only the last 10 (5) years from the last ensemble member against observations? Is that short time period really sufficient to confirm or reject the fitness of EC-Earth? I don't understand why you don't average 10-20 year from all ensemble members (start dates) and compare against the corresponding time period of the re-analysis (see also comment above about choice of re-analysis data)."*

**Our response:** The reviewer makes a very reasonable point. As implied in the manuscript, we restricted our assessment to this time period because the default CTRL version of the model was tuned for this particular time period. The reason for restricting attention to this shorter time period is a decision made by the EC-Earth Consortium as a whole, but I believe it is to avoid the possibility of 'fitting' the model's climate sensitivity. If one tries to match the energy budget over a long time period, one is effectively trying to fit the specific temperature growth, i.e. the climate sensitivity over the historical period. Fitting therefore to a very recent time period, where there is good satellite data for robust observational estimates, is therefore a pragmatic choice for model tuning.

One key conclusion drawn from this paper is that the model will in general need to be retuned after the addition of stochastic schemes, and that for this reason, comparisons of a scheme's performance against the CTRL model must always be considered with respect to the model's tuning procedure. In particular, because EC-Earth is tuned according to the historical time periods energy budget, the performance of a stochastic scheme on the energy budget over the same period will almost always look like a degradation. This is why, when comparing to observations, we only explicitly considered the short time period in question.

It is true that we could nevertheless have considered the evolution of the energy budget over the full simulation period 1960-2000. However, this essentially amounts (as explained above) to assessing the impact of the schemes on the model's climate sensitivity. This was not a topic we wanted to expand upon in this manuscript, because it is the topic of an independent manuscript currently under review (Strommen et al 2019, now cited in the revised manuscript). We have added a caveat to this effect in Section 3.4.

**9) Reviewer #2:** *"Section 4,5,6: when you talk about cloud water do you mean the gridpoint value of the in-cloud value? I would suspect it's the gridpoint value, and if that's the case then the reduction in cloud cover would imply an even larger increase in the in-cloud water content with all the consequences for optical thickness and cloud microphysics. Good if you could clarify on this point."*

**Our response:** Cloud liquid water refers to the vertical integral of liquid water contained within clouds in a single gridpoint column: we have added this descriptor to the revised manuscript in response to the reviewers point.

The point about reduced cloud cover further `exacerbating' the increased cloud water content is a good one. We have included a line to this effect in the revised manuscript, section 4.1.

**10) Reviewer #2:** *"Section 4.2,5.2,6.2 and Fig 8: I don't see a point in making a timeseries of the energy budget, the interesting aspect for a climate model is how well it simulates the average flux and its variability compared to observations (re-analysis). For that reason it would make more sense to average the biases in the fluxes and present them as a table or barplot similar to how you did in Figs 1 and2."*

**Our response:** While we agree with the reviewer that the most important aspect is the model's ability to capture the average flux and variability compared to observations, the timeseries plot does capture some relevant further information. Because the differences are effectively constant in time, this implies that the impact is not independent of the initial ocean state. Furthermore, one can infer that the model adjustment is extremely rapid, with the stochastic schemes not causing any systematic drift; because of the potentially slow response of the land-scheme, such drift cannot be ruled out a priori even in the absence of ocean coupling. Therefore, we feel there is value in keeping this as a timeseries plot.

   We do however accept that the above reasoning was not adequately included in the submitted manuscript. We have added a brief comment to this effect in sections 4.2, 5.2 and 6.2 (Energy Budget impact sections for the three schemes).

**11) Reviewer #2:** *"Section 7: what is the motivation for selecting the Hadley circulation and the QBO as test cases for the stochasticity? Why not NAO or PNA as other prominent modes of atmospheric variability?"*

**Our response:** In response to a comment by Reviewer #1, we have added motivation for these choices in the revised manuscript: see the introduction to Section 7.

**12) Reviewer #2:** *"Section 7.1 Fig 9: It is not easy to easy to distinguish the different colors of the dots (I am slightly color blind) and you should consider presenting the data in a different way, e.g. by using different symbols for the different members/time periods."*

**Our response:** We have edited Figure 9 to make the points be larger and use different symbols; we also added a legend to aid interpretation. The colourscale has been chosen due specifically to its readability for colour blind people: if the reviewer still finds the plot tricky to parse, then we would be very happy to edit it further !

**13) Reviewer #2:** *"Section 7.2: what is the reason for evaluating QBO only for the last ensemble member? Why not using the results from all start dates?"*

**Our response:** We were not clear enough in the presentation of this section. To be clearer, the estimate of the QBO period for the simulations are done using all 5 ensemble members: the period shown is the average across these 5 estimates. For Figure 11, showing a pressure-time contour plot, we only show this for the last ensemble member because the degraded QBO looks extremely similar across all 5 members. Therefore, we opted to simply show the last member as an example, against the more recent ERA-Interim reanalysis data.

We have added some clarifying remarks to this effect in Section 7.2, in response to this point. In addition, we have included errorbars in the QBO period plot, to show the spread across the 5 estimates. This illuminates the results further and immediately suggests that the full ensemble is used for the computation.

14) **Reviewer #2:** *"Section 8, 1st sentence: you cannot blame the absence of a more process-oriented analysis on the lack of available data because you have designed the experiment and its output."*

**Our response:** Yes, you are of course right. The point is rather that we made an explicit choice to construct experiments that would illuminate changes in the long-term climate, and therefore did not construct an experimental protocol that was suitable for robust assessment of the rapid response of the model. We have rephrased the introduction to Section 8 to clarify this.

We further point out there, that experiments aimed at illuminating the rapid response of the model to SPPT were carried out in Strommen et al (2019). These in fact appear to confirm the hypothesis made in the present paper, as they show that the very rapid response is to cloud liquid water and evaporation.

15) **Reviewer #2:** *"Section 8.1, p.14 l.7: it's not clear to me how an increased in-cloud liquid water content could steepen the near-surface humidity gradient. Could you explain better what you mean?"*

**Our response:** This was poorly phrased on our part. We have rephrased this part of section 8.1 in response to questions from Reviewer #1, and it is hopefully more illuminating now. For your convenience, this paragraph now reads:

"With both SPPT and ISPPT, the dominant impact on the energy budget is increased evaporation. In the IFS, the amount of evaporation at a gridpoint depends primarily on the surface wind-speeds and the extent to which the specific humidity at the surface gridpoint differs from the saturation humidity (a function of surface temperature). While wind-speeds do increase by about 1.4\% on average with ISPPT, the mean wind-speeds are unchanged with SPPT, with a tiny increase of only 0.06\%. Given that the increase in evaporation of both SPPT and ISPPT are of the same order of magnitude, this suggests changes in humidity are a key factor. Because SST's are held fixed, such changes will be, to first order, driven by changes in the water content of the atmosphere as opposed to temperature changes at the surface. One possibility is that the increase in cloud liquid water is depleting the near-surface humidity, causing more favourable conditions for evaporation. The fact that both

cloud liquid water and surface wind-speeds increase more with ISPPT could then explain why this impact is amplified in those experiments. Another possibility is that the first order impact is on convection in the tropics, which may be activated more frequently with SPPT/ISPPT. This could lead to a drying of the boundary layer, promoting more evaporation in response."

**16) Reviewer #2:** *"Section 8.2, p.15 l.13: what do you mean with runoff being a key driver of atmosphere interaction? Isn't runoff simply the difference between P-E and the amount of water absorbed by soil? It's a residual, not a driver, or?"*

**Our response:** It would indeed be more correct to say that soil moisture is the key conduit between the land and the atmosphere, through its impact on evaporative cooling and latent heat transfer. The point we are trying to make in the manuscript is that the primary impact of the LAND scheme is to affect runoff, and the changes in runoff lead to changes in soil moisture and, hence, the atmosphere as a whole.

Note that because the scheme is perturbing parameters in the soil equations themselves, it is possible for the scheme to change runoff *directly*. Note also that while *surface* runoff is effectively a residual, *subsurface* runoff in the land surface model is generated through a free drainage condition, which is dependent on soil hydrology parameters.

**17) Reviewer #2:** *"Section 8.2, p.15 l.22: why do you call runoff a tuning parameter in the LAND case? Runoff is an important diagnostics of the model that can be used to tune the model, but runoff itself isn't a tuning parameter."*

**Our response:** There seems to be a misunderstanding here. Our statement is "Tuning parameters for EC-Earth **include constants that regulate** [...] **runoff** in the LAND scheme case" (bolded for emphasis). In other words, we absolutely agree that runoff itself isn't a parameter that can be tuned, but other actual tuning parameters have a strong impact on the behaviour of runoff in the model, and these parameters *are* tuned.

We have therefore left this statement as is in the revised manuscript.

**18) Reviewer #2:** *"Code availability: EC-Earth is licensed and not openly available, it's not sufficient for any presumptive user to request access. I would suggest you check the guidelines of GMD that regulate code availability and re-phrase this section."*

**Our response:** Yes, this should have been made clearer. We did state that access was available upon requesting permission from the consortium, but should have been clearer that this is due to explicit licensing issues attached to the IFS component. We have now rephrased this.

*19)* **Reviewer #2:** *"p.2 l.12: "However" seems to be inappropriate in this sentence."*

**Our response:** We agree, and have rephrased this to "Most modern climate models also …".

*20)* **Reviewer #2:** *"p.15 l.15: shouldn't it rather be "...none of the schemes _is_ able..."?"*

**Our response:** This particular grammatical construction is one of those funny English loose ones, where I believe one can get away with using either 'is' or 'are'. However, when the meaning is `none of them', one typically uses the plural 'are'.